# Dynamic Early Exit in Reasoning Models

**Chenxu Yang**♠♡∗**, Qingyi Si**◇∗**, Yongjie Duan**◇**, Zheliang Zhu**♠♡**, Chenyu Zhu**◇**,**
**Qiaowei Li**◇**, Minghui Chen**♠♡**, Zheng Lin** ♠♡†**, Weiping Wang**♠**,**
♠Institute of Information Engineering, Chinese Academy of Sciences, Beijing, China
♡School of Cyber Security, University of Chinese Academy of Sciences, Beijing, China
◇Huawei Technologies Co., Ltd.
{yangchenxu,linzheng}@iie.ac.cn, siqingyi@huawei.com

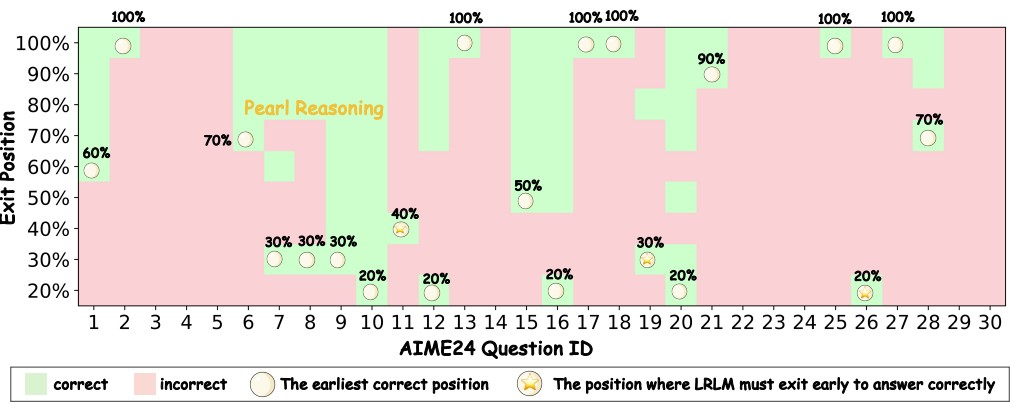

Figure 1: Correctness statistics for early exits at various reasoning steps.

## Abstract

Recent advances in large reasoning language models (LRMs) rely on test-time scaling, which extends long chain-of-thought (CoT) generation to solve complex tasks. However, overthinking in long CoT not only slows down the efficiency of problem solving, but also risks accuracy loss due to the extremely detailed or redundant reasoning steps. We propose a simple yet effective method that allows LLMs to self-truncate CoT sequences by early exit during generation. Instead of relying on fixed heuristics, the proposed method monitors model behavior at potential reasoning transition points and dynamically terminates the next reasoning chain's generation when the model exhibits high confidence in a trial answer. Our method requires no additional training and can be seamlessly integrated into existing o1-like reasoning LLMs. Experiments on 10 reasoning benchmarks (e.g., GSM8K, MATH-500, AMC, GPQA, AIME and LiveCodeBench) show that the proposed method is consistently effective on 11 cutting-edge reasoning LLMs of varying series and sizes, reducing the length of CoT sequences by an average of 19.1% to 80.1% while improving accuracy by 0.3% to 5.0%. [1]

## 1 Introduction

The emergence of large reasoning models (Xu et al., 2025a), such as DeepSeek-R1 (DeepSeek-AI et al., 2025) and GPT-O1 (OpenAI, 2025), has marked a significant breakthrough in natural language processing, particularly in solving complex and intricate tasks(WANG et al., 2025). These models leverage the test-time scaling (Snell et al., 2024) law by generating a longer CoT (Wei et al., 2023) with rich and diverse reasoning paths, unleashing the potential of their reasoning ability.

---

[*]    Equal Contribution.
[†]    Zheng Lin is the corresponding author.
[1]    The code is available at https://github.com/iie-ycx/DEER

However, the generation of overlong CoT significantly increases computational overload and reasoning latency, which hinders their deployment in computationally sensitive applications. Moreover, recent research (Chen et al., 2025b; Team et al., 2025a) reveals an intrinsic overthinking problem in LRMs: These models persistently generate verbose reasoning sequences (Wu et al., 2025; Cuadron et al., 2025) , introducing irrelevant information and unnecessary thought steps. Such redundant processing not only wastes computational resources but also leads to accuracy degradation by derailing from correct reasoning paths to erroneous ones (see Questions 11, 19 and 26 in Fig. 1. This redundancy can be attributed to the design of the Supervised Fine-Tuning (Achiam et al., 2023; Wei et al., 2021; Ouyang et al., 2022) or Reinforcement Learning (Bai et al., 2022; Ouyang et al., 2022; Schulman et al., 2017; Ramesh et al., 2024) stage, where the ability to dynamically adjust its reasoning length during generation is overlooked, leaving a gap in the inference efficiency of LRMs.

Intuitively, as the number of reasoning paths increases, more information is referenced when generating conclusions. If we can identify the critical point where the reasoning information becomes just sufficient (termed **Pearl Reasoning**) and force the model to stop further thinking and directly output conclusions at this point, we can achieve both accuracy and efficiency. This paper aims to *find such pearls in long CoT sequences*. To validate our motivation, we forced the model to switches from thinking to directly generating answers, at different transition points in the thought process. If the answers obtained are correct, the existence of such pearl reasoning is verified. As shown in Fig. 1, about 75% samples contain such pearls (early exit yields correct answers), even 36.7% samples required only less than half of the original reasoning paths to reach correct conclusions. Therefore, how to find the pearl reasoning is a valuable topic to achieve efficient reasoning.

To this end, we propose a novel, training-free approach **DEER** that allows large reasoning language models to achieve **D**ynamic **E**arly **E**xit in **R**easoning. It regards the key moments when the model switches thought chains in reasoning as chances of early exit, and prompting LRMs to stop thinking and generate trial answers at these moments. The confidence of each trial answer is the decision-making reference of early exit in reasoning. Specifically, the proposed method contains three actions: 1) **Reasoning Transition Monitoring**. During the generation of long CoTs, DEER monitors the positions of reasoning transitions through either linguistic marker-based (such as *"Wait"*) or entropy-based methods. When the reasoning transition points are found, the action of 2) **Trial Answer Inducing** is triggered: we replace it with *"final answer"* tokens to induce the model to immediately generate a trial answer, which will be used for 3) **Confidence Evaluating**. If the confidence is sufficiently high, set the model to stop further thinking and generate a conclusion based on the generated thoughts. Otherwise, the action of Trial Answer Inducing is revoked, and the model continues reasoning along the original path. Moreover, Considering the potential sensitivity of models to answer inducing prompts, we propose **DEER-Pro** (a **P**arallel and **Ro**bust variant of DEER), which performs multiple parallel answer inductions at potential early-exit points and calibrates confidence based on the aggregated results, thereby further ensuring DEER's robustness.

Our method is simple yet effective, and can be seamlessly extended to eleven reasoning models of varying architectures and sizes, achieving excellent results in the ten reasoning benchmarks, including mathematical tasks (e.g., AIME 2024, AMC 2023 and MATH-500), scientific tasks (e.g., GPQA Diamond) and programming tasks (e.g., BigCodeBench). Specifically, our method, when integrated into cutting-edge reasoning models, can reduce the length of CoT sequences by an average of 19.1% to 80.1% while improving accuracy by 0.3% to 5.0% across different reasoning benchmarks. Our DEER offers a plug-and-play solution for improving both the efficiency and accuracy of LRMs.

## 2 MOTIVATIONS AND OBSERVATIONS

In this section, we analyze the overthinking phenomenon in LRMs and investigate the impact of static early exits on model performance. We define "pearl reasoning" as the critical juncture where reasoning information becomes precisely sufficient for accurate problem-solving. Our analysis in Figure 1 reveals that approximately 75% of samples contain such pearls (where early exit yields correct answers). Furthermore, we identified a subset of samples for which correct answers are exclusively obtainable through early exits (exemplified by Questions 11, 19, and 26 in Figure 1). Quantitative analysis presented in Figure 2(a) further demonstrates that 60.8% and 35.1% of correctly answered samples in MATH-500 and GPQA, respectively, maintain their accuracy when employing early exits after completing merely 20% of the reasoning steps. These empirical findings

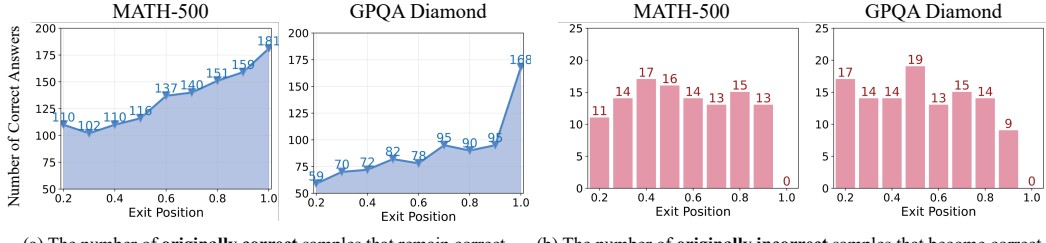

(a) The number of **originally correct** samples that remain correct with early exiting on different positions.

(b) The number of **originally incorrect** samples that become correct with early exiting on different positions.

Figure 2: Quantitative pilot experiment results. Please refer to Appendix A for setups.

substantiate our hypothesis that LRMs possess the potential to achieve simultaneous improvements in both computational efficiency and prediction accuracy through strategic early termination.

Fig. 2(b) illustrates that exiting at different positions corrects varying proportions of wrong answers. For the MATH dataset, the highest correction rate is achieved when exiting at 40% of the reasoning steps, whereas for the GPQA dataset, the optimal correction occurs when exiting at 50%. The optimal early exit point varies for each problem and is closely related to the inherent difficulty of the problem itself. Therefore, it is intuitive that relying on a static early exit strategy based on fixed heuristics is suboptimal, underscoring the necessity of designing a dynamic early exit mechanism.

## 3 METHOD

### 3.1 THE GENERATION PATTERN OF LARGE REASONING MODELS

In contrast to traditional large language models (*System 1*), large reasoning models (*System 2*) (Li et al., 2025b) exhibit distinct generation patterns during the inference stage. (1) LRMs use delimiters to divide the output into two processes: slow thinking and conclusion. LRMs conduct systematic and thorough reasoning in the slow thinking, ultimately summarizing the thought process and providing the final answer in the conclusion. (2) During the slow thinking process, LRMs engage in complex thinking actions (thoughts), such as problem comprehension, approach exploration, and result verification (Luo et al., 2025b). Within each reasoning action (thought), the model performs specific procedural action execution, while transitions between different reasoning actions are typically marked by action transition points (**ATP**), such as *"Wait"*, *"Alternatively"*.

$$\textit{System 1: } [\text{Prompt}] + [\text{Completion}], \tag{1}$$

$$\textit{System 2: } [\text{Prompt}] + \langle\text{think}\rangle + [\text{Slow Thinking}] + \langle\text{/think}\rangle + [\text{Conclusion}], \tag{2}$$

$$[\text{Slow Thinking}] : [\text{Action Execution}] + (\textbf{ATP}) + [\text{Action Execution}] + (\textbf{ATP}) + \cdots, \tag{3}$$

where $\langle\text{think}\rangle$ and $\langle\text{/think}\rangle$ are begin-of-thinking and end-of-thinking delimiters respectively.

### 3.2 DYNAMIC EARLY EXIT IN REASONING

In this section, we introduce the Dynamic Early Exit in Reasoning (DEER) method to determine optimal positions for early exits (pearl reasoning path), thereby alleviating the overthinking issue.

The core idea behind DEER is that a model's confidence in its trial answer dynamically indicates whether the thinking information required for LRMs to generate the final answer is sufficient. We observe that when the model's reasoning process is incomplete or flawed, the trial answer tends to exhibit significantly lower confidence. Conversely, when the reasoning is comprehensive and logically sound, the model generates answers with higher confidence, as illustrated in Fig. 17. This suggests that the model implicitly recognizes when **pearl reasoning** occurs, but lacks an explicit mechanism during inference to leverage this awareness for early termination. DEER aims to bridge this gap by converting implicit awareness into explicit early-exit decisions.

As shown in Fig. 3, DEER involves three designs to determine whether to exit early: reasoning transition monitor, answer inducer, and confidence evaluator.

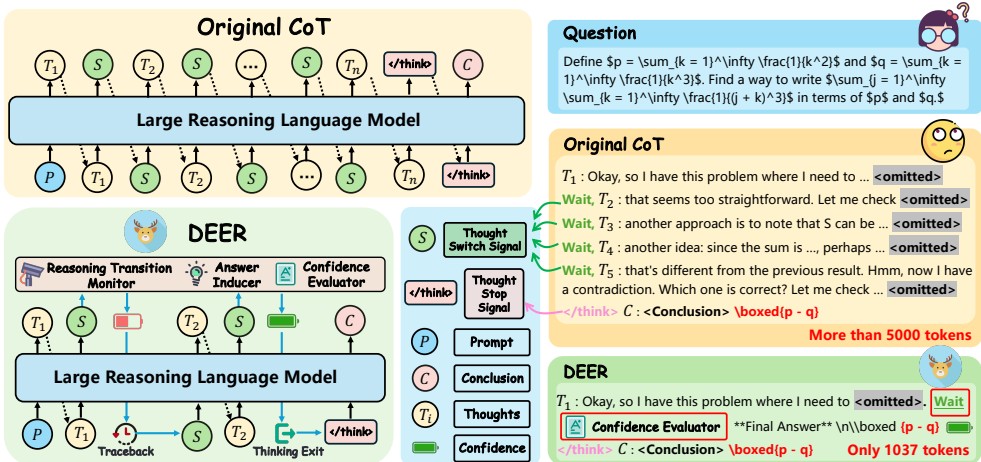

Figure 3: An overview of the Dynamic Early Exit in Reasoning (DEER) method.

**Reasoning transition monitor.** Within the DEER framework, we propose two alternative monitor design strategies: (i) **linguistic marker-based**, and (ii) **entropy-based** monitoring. For the first strategy, as mentioned in Section 3.1, LRMs explicitly utilize ATPs to mark boundaries between different thoughts. This feature enables DEER to recognize ATPs as potential early-exit opportunities. In the second strategy, DEER employs "$\backslash n \backslash n$" as delimiters to demarcate reasoning steps. Following each reasoning step, DEER computes the entropy of the initial token, denoted as $H(p(\cdot|x_{<t}))$. Low entropy values indicate that the model is engaged in procedural action execution, characterized by stable reasoning patterns. Conversely, high entropy values suggest that the model is deliberating on its subsequent reasoning action, with multiple potential pathways being activated concurrently. These positions exhibiting high entropy are identified as candidate points for early-exit.

Our subsequent experiments in Section 4.3 demonstrate that DEER with external linguistic markers satisfies similar properties as the second internal state-based approach while achieving comparable performance. When applied to English LRMs, existing models consistently exhibit a pattern of generating such linguistic markers. Following the Occam's Razor principle (Rasmussen & Ghahramani, 2000), we recommend adopting the first strategy. For non-English reasoning scenarios, the alternative second strategy can also accurately capture early-exit points, demonstrating the generality and robustness of DEER.

**Answer inducer.** When the LRM pauses at a potential early exit point, the trial answer inducer module prompts the model to generate an intermediate answer based on the reasoning content produced so far. We incorporated the answer delimiters ($\backslash boxed\{\}$) into the prompt to facilitate a more precise identification of the trial answers, as follows: $\boldsymbol{A} = \text{LRM}(\boldsymbol{P}, \boldsymbol{T}, \boldsymbol{I})$ where $\boldsymbol{P}$ denotes the input prompt, $\boldsymbol{T}$ denotes the generated thoughts, $\boldsymbol{I}$ denotes the answer inducer prompt, and $\boldsymbol{A} = [a_0, a_1, \ldots, a_n]$ is the trial answer.

**Confidence evaluator.** The confidence evaluator computes the confidence of the induced trial answer. It takes the maximum predicted probability of each token as its confidence. For multi-token trial answers, the overall confidence is computed as their mean score across all tokens as follows:

$$p(a_t) = \text{softmax}(\mathcal{M}(\boldsymbol{P}, \boldsymbol{T}, \boldsymbol{I}, \boldsymbol{a}_{<t})), \quad \mathcal{C} = \left(\prod_{i=1}^{n} \max_{a_t \in \mathcal{V}} p(a_t)\right)^{1/n} \quad (4)$$

where $\mathcal{M}$ is the language model head at the final layer of the LRM. The calculation of $\mathcal{C}$ employs the geometric mean, which better aligns with the multiplicative nature of joint probabilities and exhibits greater sensitivity to low probability values, thereby providing enhanced robustness.

Finally, the comparison between the obtained confidence and the empirical threshold $\lambda$ determines whether to exit early. If $\mathcal{C} > \lambda$, we consider the reasoning information currently generated by the LRM to be sufficient, indicating that the model has reached the **pearl reasoning**. At this point, DEER stops further reasoning actions and proceeds to deliver the conclusion. Otherwise, the model reverts to the previous transition point to generate the next thoughts.

**DEER-PRo.** To further improve the reliability and accuracy of pearl reasoning identification, we introduce **DEER-PRo**, a **P**arallel and **Ro**bust variant of DEER. Through answer elicitation using varied prompts at early-exit points, DEER-PRo calculates both the mean and Mean Absolute Deviation (MAD) of multiple confidence values, deriving a calibrated confidence score $\mathcal{C}_{cali}$ as follows:

$$\mathcal{C}_{cali} = \mathcal{C}_{avg} - \alpha \cdot \mathcal{C}_{\text{MAD}}, \quad \mathcal{C}_{avg} = \frac{1}{N}\sum_{i=1}^{N}\mathcal{C}_i, \quad \mathcal{C}_{\text{MAD}} = \frac{1}{N}\sum_{i=1}^{N}|\mathcal{C}_i - \mathcal{C}_{avg}| \tag{5}$$

where $\mathcal{C}_i = \left(\prod_{i=1}^{n}\max_{a_t \in \mathcal{V}} \text{softmax}(\mathcal{M}(\boldsymbol{P}, \boldsymbol{T}, \boldsymbol{I}_i, \boldsymbol{a}_{<t}^i))\right)^{1/n}$ denotes the confidence score obtained using a specific answer inducing prompt $\boldsymbol{I}_i$, $N$ denotes the number of inducing attempts, and $\alpha$ is the fluctuation penalty strength coefficient. The introduced conservative bias $\mathcal{C}_{\text{MAD}}$ effectively prevents erroneous early exits caused by overestimated confidence scores resulting from positive noise in prompts, where the estimated confidence exceeds the true confidence. We demonstrate in the Appendix B that $\mathcal{C}_{cali}$ effectively eliminates the influence of the model's sensitivity to answer inducer prompts on early-exit accuracy, thus substantially improving DEER's robustness.

## 3.3 BRANCH-PARALLEL DECODING ACCELERATION

Intuitively, the computation of the answer inducer and confidence evaluator in DEER introduces additional latency during inference, particularly in code generation tasks where trial answers remain lengthy. This overhead diminishes the efficiency gains achieved through substantial reduction of generated CoT sequences. To address this challenge, we integrate DEER with a branch-parallel acceleration strategy (Fig. 9) that mitigates these efficiency limitations through two key mechanisms: (1) linearization of multiple branches into a single sequence for parallel generation using a specialized causal attention mask, and (2) dynamic KV cache management via confidence-based pruning. This strategy facilitates temporal overlap between trial answer evaluation and concurrent reasoning-chain generation, thereby optimizing overall inference efficiency.

## 4 EXPERIMENTS

### 4.1 EXPERIMENTAL SETUP

**Benchmarks, Metrics and Implementations.** We evaluate model performance across 10 benchmarks, including 6 mathematical reasoning benchmarks: GSM8K (Cobbe et al., 2021), MATH-500 (Hendrycks et al., 2021), AMC 2023 (AI-MO, 2024), AIME 2024, AIME 2025 (MAA Committees), OlympiadBench (He et al., 2024), one scientific reasoning benchmark: GPQA Diamond (Rein et al., 2023), and 3 code reasoning benchmarks: HumanEval (Chen et al., 2021), BigCodeBench (Zhuo et al., 2024), and LiveCodeBench (Jain et al., 2024). Among the six mathematical reasoning benchmarks, GSM8K, MATH-500, and AMC 2023 are generally considered to be relatively simple reasoning tasks, whereas AIME 2024, AIME 2025, and OlympiadBench are regarded as more challenging. Given the extensive set of evaluation benchmarks, we selectively present the most popular ones (GSM8K, MATH-500, AMC 2023, AIME 2024 and GPQA Diamond) in the main experiment. More experimental results are provided in the Appendix K. We selected *Accuracy* (**Acc**), *Token Number* (**Tok**), and *Compression Rate* (**CR**) as the evaluation metrics. **Acc** denotes the final answer accuracy. **Tok** denotes the average generation length per sample to evaluate the cost. **CR** is defined as the ratio of the average response length to that of the original model, with lower values indicating higher compression. Given the limited number of samples in datasets AMC 2023, AIME 2024, and AIME 2025, we conduct 4 sampling rounds per instance and average the results across all metrics to ensure stability and reliability. We have implemented DEER using both HuggingFace Transformers (Wolf et al., 2020) and the vLLM inference acceleration framework (Kwon et al., 2023). The experimental results presented in this paper are based on the vLLM implementation. We set the hyperparameter $\lambda$ to 0.95 ($\lambda = 0.95$). For entropy-based DEER, following the 80/20 principle proposed in (Wang et al., 2025b), we designate reasoning step termination positions with entropy values exceeding 0.672 as early-exit points. For DEER-Pro, we set $N = 4$ and $\alpha = 1$. More experimental setup details are placed in Appendix C.

**Backbone LRMs and Baselines.** We conducted experiments on the open-source DeepSeek-R1-Distill-Qwen series of models (1.5B, 7B, 14B, and 32B)(DeepSeek-AI et al., 2025), Qwen3 series of

Table 1: Experimental results across various types of reasoning models. "Acc" denotes accuracy, "Tok" denotes token count, and "CR" denotes compression rate. ↑ indicates that higher values are better, while ↓ indicates that lower values are better. The best results are highlighted in **bold**.

| | | MATH | | | | | | | | | | | | | SCIENCE | | Overall | |
|---|---|---|---|---|---|---|---|---|---|---|---|---|---|---|---|---|---|---|
| | GSM8K | | | MATH-500 | | | AMC23 | | | AIME24 | | | | GPQA-D | | | | |
| Method | Acc↑ | Tok↓ | CR↓ | Acc↑ | Tok↓ | CR↓ | Acc↑ | Tok↓ | CR↓ | Acc↑ | Tok↓ | CR↓ | | Acc↑ | Tok↓ | CR↓ | Acc↑ | CR↓ |
| *DeepSeek-R1-Distill-Qwen-7B* | | | | | | | | | | | | | | | | | | |
| *Vanilla* | 89.6 | 1,484 | 100% | 87.4 | 3,858 | 100% | 78.8 | 6,792 | 100% | 41.7 | 13,765 | 100% | | 23.7 | 10,247 | 100% | 64.2 | 100% |
| *TCC* | 88.0 | 892 | 60.1% | 89.2 | 3,864 | 100.2% | 82.5 | 6,491 | 95.6% | 48.4 | 10,603 | 77.0% | | 27.3 | 8,442 | 82.4% | 67.1 | 83.0% |
| *CoD* | 84.7 | 298 | 20.1% | 83.2 | 1,987 | 51.5% | 77.5 | 4,440 | 65.4% | 40.0 | 10,519 | 76.4% | | 37.9 | 6,431 | 62.8% | 64.7 | 55.3% |
| *NoThinking* | 87.1 | 284 | 19.1% | 80.6 | 834 | 21.6% | 65.0 | 1,911 | 28.1% | 26.7 | 4,427 | 32.2% | | 29.8 | 724 | 7.1% | 57.8 | **21.6%** |
| *Dynasor-CoT* | 89.6 | 1,285 | 86.6% | 89.0 | 2,971 | 77.0% | 85.0 | 5,980 | 88.0% | 46.7 | 12,695 | 92.2% | | 30.5 | 7,639 | 74.5% | 68.2 | 83.7% |
| *SEAL* | 88.4 | 811 | 54.6% | 89.4 | 2,661 | 69.0% | – | – | – | – | – | – | | – | – | – | – | – |
| *DEER* | 90.6 | 917 | 61.8% | 89.8 | 2,143 | 55.5% | 85.0 | 4,451 | 65.5% | 49.2 | 9,839 | 71.5% | | 31.3 | 5,469 | 53.4% | 69.2 | 61.5% |
| *DEER-PRo* | 91.0 | 989 | 66.7% | 90.2 | 2,391 | 62.0% | 87.5 | 4,877 | 71.8% | 49.2 | 10,046 | 73.0% | | 30.6 | 5,682 | 55.5% | **69.7** | 65.8% |
| *Qwen3-14B* | | | | | | | | | | | | | | | | | | |
| *Vanilla* | 95.1 | 2,047 | 100% | 93.8 | 4,508 | 100% | 95.0 | 7,139 | 100% | 70.0 | 10,859 | 100% | | 60.1 | 7,339 | 100% | 82.8 | 100% |
| *TCC* | 95.7 | 1,241 | 60.6% | 94.6 | 4,484 | 99.5% | 95.0 | 7,261 | 101.7% | 70.8 | 11,573 | 106.6% | | 60.1 | 7,138 | 97.3% | 83.3 | 93.1% |
| *CoD* | 85.7 | 648 | 31.7% | 75.2 | 2,359 | 52.3% | 72.5 | 4,122 | 57.7% | 60.0 | 10,768 | 99.2% | | 51.0 | 1,177 | 16.0% | 68.9 | 51.4% |
| *NoThinking* | 94.8 | 286 | 14.0% | 85.0 | 1,228 | 27.2% | 77.5 | 2,133 | 29.9% | 26.7 | 7,337 | 67.6% | | 50.5 | 2,320 | 31.6% | 66.9 | **34.1%** |
| *Dynasor-CoT* | 95.6 | 1,483 | 72.4% | 93.8 | 4,063 | 90.1% | 95.6 | 6,582 | 92.2% | 73.3 | 10,369 | 95.5% | | 59.6 | 5,968 | 81.3% | 83.6 | 86.3% |
| *DEER* | 95.3 | 840 | 41.0% | 93.0 | 3,074 | 68.2% | 95.0 | 4,763 | 66.7% | 76.7 | 7,619 | 70.2% | | 57.6 | 2,898 | 39.5% | 83.7 | 57.1% |
| *DEER-PRo* | 95.3 | 926 | 45.2% | 94.4 | 3,260 | 72.3% | 95.6 | 4,905 | 68.7% | 75.0 | 8,135 | 74.9% | | 61.2 | 4,062 | 55.4% | **84.3** | 63.3% |
| *QwQ-32B* | | | | | | | | | | | | | | | | | | |
| *Vanilla* | 96.7 | 1,427 | 100% | 93.8 | 4,508 | 100% | 92.5 | 6,792 | 100% | 66.7 | 10,821 | 100% | | 63.1 | 7,320 | 100% | 82.6 | 100% |
| *TCC* | 95.8 | 1,348 | 94.5% | 94.4 | 4,315 | 95.7% | 90.0 | 6,818 | 100.4% | 60.0 | 11,263 | 104.1% | | 61.6 | 7,593 | 103.7% | 80.4 | 99.7% |
| *CoD* | 96.0 | 627 | 43.9% | 94.0 | 3,630 | 80.5% | 92.5 | 5,943 | 87.5% | 60.0 | 10,731 | 99.2% | | 62.6 | 6,039 | 82.5% | 81.0 | **78.7%** |
| *NoThinking* | 96.2 | 1,113 | 78.0% | 94.8 | 3,930 | 87.2% | 87.5 | 6,908 | 101.7% | 66.7 | 10,859 | 100.4% | | 63.6 | 7,668 | 104.8% | 81.8 | 94.4% |
| *Dynasor-CoT* | 95.2 | 1,095 | 76.7% | 94.2 | 4,176 | 92.6% | 93.8 | 6,544 | 96.3% | 63.3 | 11,156 | 103.1% | | 64.1 | 7,024 | 96.0% | 82.1 | 93.0% |
| *DEER* | 96.3 | 977 | 68.5% | 94.6 | 3,316 | 73.6% | 95.0 | 5,782 | 85.1% | 70.0 | 10,097 | 93.3% | | 64.1 | 6,163 | 84.2% | 84.0 | 80.9% |
| *DEER-PRo* | 96.2 | 1032 | 72.3% | 94.8 | 3,650 | 80.9% | 95.0 | 5,811 | 85.6% | 70.0 | 10,264 | 94.9% | | 64.7 | 6,201 | 84.7% | **84.1** | 83.7% |

models (1.7B, 4B, 8B, 14B, 32B) (Qwen et al., 2025), QwQ-32B (Team, 2025), and DeepSeek-R1 (Liu et al., 2025c). Due to the large number of models evaluated, we selectively present DeepSeek-R1-Distill-Qwen-7B, Qwen3-14B, and QwQ-32B as representative examples in the main experiment. More experimental results are provided in the Appendix C. We compare DEER against existing prompt-based and output-based efficient reasoning approaches, including *Vanilla*, *TCC* (Muennighoff et al., 2025), *CoD* (Xu et al., 2025c), *NoThinking* (Ma et al., 2025a), *Dynasor-CoT* (Fu et al., 2025), and *SEAL* (Chen et al., 2025a). *Vanilla* performs direct evaluation of the LRM without any intervention. Token-Conditional Control (*TCC*) specifies a fixed token count in the system prompt to enforce a token budget; in our experiments, we set this limit based on the actual token length generated by DEER. Chain-of-Draft (*CoD*) reduce verbosity by limiting the number of words used in each reasoning step, focusing only on the essential calculations or transformations needed to progress. *NoThinking* prompts the model to skip the reasoning phase and directly generate the final answer. *Dynasor-CoT* periodically prompts the model to produce intermediate answers at fixed token intervals and triggers early exit when three consecutive answers are consistent. *SEAL* trains a steering vector to calibrate the CoT process, guiding the model toward more reliable reasoning.

## 4.2 MAIN RESULTS

**Overall Performance.** Due to space constraints, Tab.1 presents five widely adopted reasoning benchmarks, evaluated across three state-of-the-art reasoning models specifically covering three model scales, which comprehensively demonstrates DEER's superior performance. We also provide more results across 10 datasets covering 11 models ranging from 1.5B to 671B parameters in the Appendix. It can be found that DEER demonstrates strong adaptability across various reasoning models and tasks, achieving accuracy improvements of 0.9 to 4.8 points while reducing sequence length by 19.1% to 42.9% compared to vanilla models. DEER-Pro achieves higher accuracy with only a marginal increase in generation length ranging from 2.8% to 6.2% compared to DEER. We conducted comparative experiments between DEER and DEER-Pro on additional smaller-scale models. The experimental results in Table 2 demonstrate that DEER-Pro achieves more significant accuracy improvements. It indicates that DEER-Pro effectively addresses the prompt sensitivity issues in smaller models, demonstrating its superior robustness.

**Comparison with Efficient Reasoning SoTAs.** Tab. 1 presents comparisons between DEER and recent efficient reasoning methods. It can be observed that DEER consistently outperforms all baselines, whereas baselines either struggle to generalize across tasks and base models, or must trade off accuracy for efficiency. Specifically, while TCC Muennighoff et al. (2025) achieve reasonable

Figure 4: Experimental results of DEER compared to Vanilla CoT across DeepSeek-R1-Distill-Qwen-Series models of varying sizes on MATH-500 and AIME 2024.

efficiency-accuracy tradeoffs on simpler tasks like GSM8K by incorporating token budgets into prompts, it fails on complex problems (such as AIME24) where models ignore prompts' length constraints and generate even longer responses than vanilla CoT. As for *NoThinking* and *CoD*, while achieving dramatic length reduction, they severely compromises models' inherent reasoning capabilities. In contrast, Dynasor-CoT preserves reasoning quality but suffers from late termination due to its conservative early-exit condition, resulting in minimal length reduction. Notably, nearly all baselines fail completely on QwQ-32B due to the sporadic invalidation of its end-of-thinking delimiter `</think>` where the model continues generating reasoning steps after it and often produces duplicate `</think>` tokens (as shown in Appendix Fig. 18). Remarkably, DEER still achieves a 19.1% length reduction on QwQ-32B despite these challenges, further demonstrating its robustness.

**Performance on Programming Tasks.** Tab. 3 reports DEER's evaluation results across three programming tasks, completing our comprehensive coverage of reasoning models' three primary domains: mathematics, science, and programming. It demonstrates DEER's consistent effectiveness across varying programming tasks and model sizes, achieving smaller compression ratios compared to math and science tasks (average 19.9% vs. 61.5%). This enhanced compression likely originates from the inherent characteristics of code generation, where each reasoning step typically produces verbose code segments containing substantial redundant tokens.

**Performance Trends across Model Sizes and Reasoning Difficulty.** Fig.4, 10 presents evaluation results on MATH-500 and AIME 2024 datasets to examine DEER's performance gains across different model sizes. It can be seen that DEER consistently enhances accuracy while reducing token consumption across all model sizes. A key observation is that smaller models (e.g., 1.5B) tend to generate significantly longer reasoning sequences with more severe overthinking phenomena. This stems from their limited reasoning capacity in discovering the correct reasoning steps during CoT generation. Consequently, our method achieves greater length reduction for these smaller models. Fig. 4 utilizes the MATH-500 (simple reasoning) and AIME 2024 (challenging reasoning) datasets as representative benchmarks. The results demonstrate DEER's dual capability: it achieves more superior compression ratios on simpler problems while delivering more substantial accuracy gains on complex tasks. This precisely addresses two critical needs in reasoning systems: the efficiency demands in simple scenarios and the growing accuracy requirements in challenging scenarios.

## 4.3 ABLATION STUDY

**Performance Trends across Token Budges.** Fig. 5 evaluates DEER's performance across varying token budgets (controlled by different max length settings). In plots (a) and (f), the x-axis represents the actual length of model-generated CoT sequences, while the y-axis indicates model accuracy. The optimal balance between accuracy and efficiency is demonstrated by curves positioned closer to the top-left corner. The blue shaded regions quantitatively represent DEER's performance gains: vertical height corresponds to accuracy improvement and horizontal width to token compression benefit. It can be seen that DEER consistently outperforms vanilla methods, as all points located upper-left to vanilla ones. As shown in the four-column plots on the right, we observe that vanilla models generate longer sequences with higher accuracy as token budgets increase, confirming test-time scaling. Notably, DEER demonstrates an adaptive tradeoff: under constrained token budgets, it achieves greater gains in accuracy but reduced benefits in length compression. Conversely, the opposite trend is observed with larger token budgets. This indicates that our method can dynamically adjust token budgets to meet varying requirements for accuracy-efficiency in different scenarios.

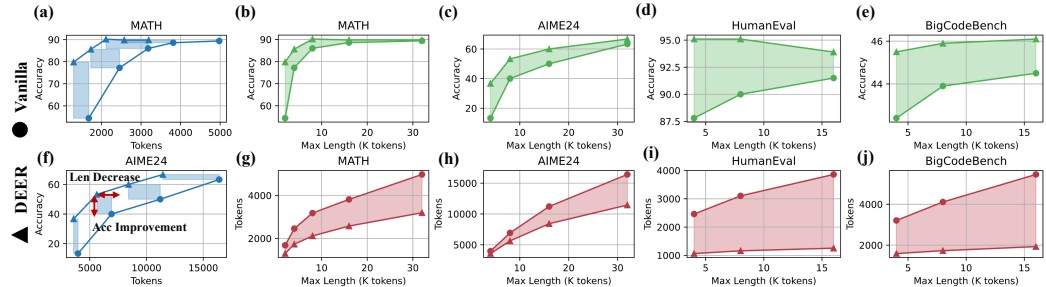

Figure 5: Performance comparison between DEER and baselines based on the DeepSeek-R1-Distill-Qwen-14B model across four datasets under different token budget settings.

**Impact of Reasoning Transition Monitor Choices.** In the main experiments, we employ *"Wait"* as the early-exit monitoring signal, denoted as DEER(W). This simple linguistic marker-based approach yields promising results. To compare the impact of different early-exit signals on DEER performance, we conduct additional experiments using *"Alternatively"* as the signal, as well as entropy-based monitoring for early-exit detection. The corresponding results are presented in Tab. 5 and 6. Tab. 5 collects statistics on the number and average length of reasoning chunks obtained by dividing the original CoT with potential exit points. The chunk numbers indicate that DEER(Ent) presents the most early-exit opportunities while DEER(A) offers the fewest, exhibiting a negative correlation with average generation length. The results in Tab. 6 across additional datasets and models demonstrate that both entropy-based and linguistic marker-based monitoring exhibit comparable superior performance, significantly outperforming the baseline. In large-scale real-world deployments, we advocate for the linguistic marker-based approach given its implementation simplicity and efficiency. Appendix I provides further exploration between these two monitoring strategies.

**Robustness of threshold $\lambda$.** Fig. 6 shows the performance of DEER on MATH-500 dataset with different threshold $\lambda$. The results indicate that when the threshold is set too low, a minor additional reduction in reasoning length leads to a significant drop in accuracy, reflecting an overcorrection of overthinking. Conversely, when the threshold is set too high, the model exits reasoning too late, resulting in prolonged reasoning lengths with a decline in accuracy. Moreover, it can be seen that our method is robust to $\lambda$ whthin the range of 0.9-0.97, eliminating the need for hyperparameter tuning. Tab. 4 presents our robustness investigation of the threshold $\lambda$ across additional datasets using Qwen3. The experimental results demonstrate that Qwen3 ex-

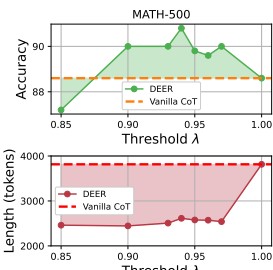

Figure 6: Impact of $\lambda$.

hibits superior robustness to $\lambda$, maintaining consistently strong performance within the range of 0.8-0.97. Additionally, the experimental results in the appendix, conducted across 11 models and 10 datasets, uniformly employ 0.95 as the threshold value. The consistently strong results further demonstrate DEER's generalization capability and robustness. Appendix Section J reveals that the underlying source of DEER's robustness originates from confidence polarization phenomenon.

## 4.4 DISCUSSION

**Efficiency Improvement.** To accurately verify the gains brought by DEER and its Branch-Parallel accelerated variant in end-to-end inference efficiency, we measured the average latency on MATH-500 and AMC 2023 datasets based on huggingface transformers (Wolf et al., 2020). As shown in Fig. 7 (a), original DEER reduces the latency by 27.9% to 40.1% while the proposed branch-parallel decoding variant reduces the latency by 36.3% to 58.6%. This suggests that Branch-Parallel DEER achieves further speed improvements by efficiently reducing

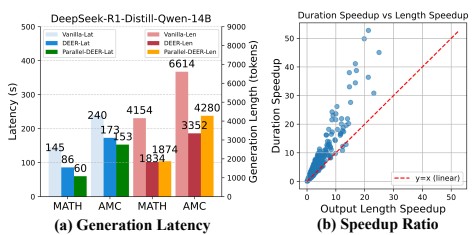

Figure 7: Efficiency Improvement.

the latency of trial answer inducing and confidence evaluation. Additionally, we mapped the latency speedup against the length savings for every sample on MATH-500. Fig. 7 (b) illustrates that the

Figure 8: Early-exit rates and accuracy of early-exited samples of DEER. Figures (a) and (b) share a common legend, as do figures (c) and (d). The height of each bar reflects the average value.

ratio between latency speedup and length savings exhibits a superlinear trend, reinforcing the significance of DEER in enhancing inference speed. In Section H of the Appendix, we theoretically prove the efficiency of DEER and explain the underlying cause of the superlinear speedup.

**Exploring the Effectiveness of DEER's Early-Exit Mechanism.** Fig. 8 presents the early-exit rate and the accuracy of early-exited samples across Qwen3-series models of varying sizes. As shown in Fig. 8(a), DEER's early-exit rate decreases with increasing task difficulty, which accounts for its relatively lower compression performance on complex tasks compared to simpler ones. Fig. 8(b) reveals that, although early-exit accuracy declines somewhat as task difficulty increases, it remains consistently high—ranging from 88% to 98%. In a concurrent study, Zhang et al. (2025a) trained a probe to decide whether to early exit. However, their probe achieves an accuracy of only around 80% on MATH-500, which is significantly lower than DEER's 95%. This indicates that the model inherently possesses the ability to assess answer correctness, and that DEER effectively harnesses this capability. As illustrated in Figures 8(c) and (d), although minor differences exist across tasks of varying difficulty, there is a general trend toward higher early-exit rates and improved accuracy as model size increases. This observation implies a positive relationship between DEER's performance and the capacity of the model: larger models yield more accurate confidence estimates, which in turn lead to better early-exit decisions. We further investigate the reasons behind DEER's accuracy gains. Fig. 11 indicates that DEER corrects more answers (green bars) than it alters incorrectly (red bars) through early exits. This suggests that DEER not only saves computational cost by exiting early on questions it could correctly answer, but also corrects problematic thinking.

**Case Study.** Fig. 13 shows that both DEER and vanilla CoT arrive at the correct answer during the first reasoning step, as shown in the green box. The difference lies in the fact that DEER exits early after evaluating the confidence of the trial answer as sufficiently high, thus producing the correct result. In contrast, the vanilla CoT proceeds to the next reasoning action. After double-checking and switching reasoning approaches, the model becomes trapped in an endless cycle of verification due to inconsistent answers from the two approaches, ultimately failing to provide a final answer. Besides, Fig. 17 shows that LRMs implicitly know when to leave early, and our method is simple and effective to realize such potential of the model itself. Please refer to Appendix L for details.

## 5 RELATED WORK

Following the taxonomy of efficient reasoning established in (Sui et al., 2025; Wang et al., 2025a), we categorize related work into three classes: **post-traning based** methods use SFT (Yu et al., 2024; Kang et al., 2025; Xia et al., 2025; Ma et al., 2025b; Munkhbat et al., 2025; Liu et al., 2024; Han et al., 2024) with variable-length CoT data or incorporate length rewards (Team et al., 2025b; Luo et al., 2025a; Aggarwal & Welleck, 2025; Arora & Zanette, 2025; Yeo et al., 2025; Shen et al., 2025b; Qu et al., 2025; Cui et al., 2025) in reinforcement learning to enable the model to adaptively generate chains of thought of different lengths, which is beyond our training-free scope. **Prompt-based** methods (Han et al., 2024; Xu et al., 2025b; Lee et al., 2025; Renze & Guven, 2024; Chen et al., 2024) use varying prompts to enforce reasoning models to generate concise CoT with less unnecessary reasoning steps. **Output-based** methods aim to accelerate reasoning generation during the model's decoding phase, and DEER falls into this category. However, most prior works (Xie et al., 2023; Liao et al., 2025; Li et al., 2024; Manvi et al., 2024; Aggarwal et al., 2023) focus on optimizing best-of-N sampling, which is irrelevant to our study. Instead, we select three recent concurrent works Nothinking (Ma et al., 2025a), Dynasor-CoT (Fu et al., 2025), and SEAL (Chen et al., 2025a) as baselines for comparison. More related works can be seen in Appendix M.

# 6 CONCLUSION

This paper verifies the rationale behind the early exit motivation in CoT generation, and accordingly proposes a training-free dynamic early exit algorithm, which makes the reasoning model withdraw from subsequent thinking when the thinking amount is just enough. Our method comprehensively evaluated across reasoning models of varying model sizes and demonstrates superior performance with fewer tokens on ten classical reasoning benchmarks, which offers a win-win solution to the trade-off between accuracy and efficiency commonly encountered in test-time scaling.

# 7 ETHICS STATEMENT

This work adheres to the ICLR Code of Ethics. We affirm that our research has been conducted with integrity, honesty, and respect for ethical principles throughout all stages of the work.

- All findings presented in this paper are reported accurately and honestly. We have not fabricated, falsified, or misrepresented any data or results. Our methods and experimental procedures are described transparently to ensure reproducibility.

- All datasets used in this research were obtained and utilized in accordance with their licenses and terms of use. For any data involving personal information, we ensured compliance with privacy regulations and obtained appropriate ethical approvals where necessary.

- All contributions to this work have been properly acknowledged. We have appropriately cited all sources and prior work that influenced our research. All co-authors have made substantial contributions to the work and have agreed to the submission.

- We have carefully considered the broader implications of our work. While our research aims to advance the field positively, we acknowledge potential dual-use concerns and encourage responsible deployment of our methods in real-world applications.

# 8 REPRODUCIBILITY STATEMENT

To ensure the reproducibility of our results, we have provided detailed descriptions of our experimental setup, hyperparameters, and implementation details. Code and supplementary materials are made available where possible to facilitate verification and future research.

# 9 ACKNOWLEDGMENTS

This work was supported by the National Natural Science Foundation of China (Nos. 62472419, 62472420) and the Enterprise Project (No. E4V06811F3).

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

## A  PILOT EXPERIMENT SETUP

We selected AIME2024 (MAA Committees) as the test set for exploratory experiments to perform a qualitative analysis and further conducted a quantitative analysis through experiments on MATH-500 (Hendrycks et al., 2021), GPQA-Diamond (Rein et al., 2023). All experiments were conducted on DeepSeek-R1-Distill-Qwen-14B (DeepSeek-AI et al., 2025). In our experiments, we first enabled the LRM to perform a complete inference on the test set (including both the slow thinking and conclusion contents). Then, we preserved the thinking content and divided it into thinking chunks based on the action transition points. Samples with more than five thinking chunks were retained. For these samples, we retained varying proportions (20%-90%) of their thinking chunks and appended an end-of-thinking token delimiter to each truncated reasoning sequence to forcibly terminate the slow-thinking process. The model then generated its final conclusion based on the partial reasoning contents. For the conclusions obtained with varying thinking contents, we evaluated their correctness and presented the results of each sample in Figure 1. Furthermore, we investigated the number of samples that remained correct after early exiting when they were originally correct, as well as the number of samples that became correct after early exiting when they were originally incorrect, across three datasets in Figure 2.

## B  PROOF OF DEER-PRO'S EFFECTIVENESS AGAINST NOISE.

### B.1  NOISE INDEPENDENCE OF THE MAD-CALIBRATED STRATEGY

Let us define the true confidence $\mu$ as the model's actual probability of deriving the correct answer given the current reasoning content, corresponding to the real probability of pearl reasoning existing at this location. The early-exit decision threshold is denoted as $\lambda$, and the model executes early exit when $\mu > \lambda$. Since answer inducing prompts may introduce $\varepsilon$, the model's output confidence fluctuates around the true confidence, yielding an observed confidence of $\mathcal{C}_i = \mu + \varepsilon_i$. In practical testing environments, we compare $\mathcal{C}_i$ with $\lambda$ to determine whether to perform early exit. Without loss of generality, we assume the noise terms $\varepsilon_i$ to be independently and identically distributed (i.i.d.), a Gaussian distribution with mean 0 and standard deviation $\sigma$, i.e., $\varepsilon_i \sim N(0, \sigma^2)$. Here, $\sigma$ represents the model's sensitivity to prompt phrasing. A larger $\sigma$ indicates higher model sensitivity, resulting in greater confidence fluctuations.

Next, we demonstrate DEER-PRO's effectiveness against noise interference by comparing the decision error rates of DEER-PRO ($\mathcal{C}_{cali}$), DEER ($\mathcal{C}_i$), an averaging approach ($\mathcal{C}_{avg}$) in critical risk scenarios. Suppose $\mathcal{C}_{cali}$ and $\mathcal{C}_{avg}$ each conduct $N$ times answer inducing in parallel, with $N$ identical to that in Equation (5). Given that preserving accuracy takes precedence over early-exit speedup gains in reasoning scenarios, we designate risk scenarios as those where true confidence $\mu < \lambda$. We then compute the false positive probability of observing $\mathcal{C} > \lambda$ due to noise interference.

### B.1.1  PROBABILITY OF ERROR FOR A SINGLE PROMPT:

$$P_{\text{FP}}(\text{Single}) = P(\mathcal{C}_i > \lambda) = P(\mu + \varepsilon_i > \lambda) = P(\varepsilon_i > \lambda - \mu) \tag{6}$$

Since $\varepsilon_i \sim N(0, \sigma^2)$, we can standardize it as:

$$P_{\text{FP}}(\text{Single}) = P\left(\frac{\varepsilon_i}{\sigma} > \frac{\lambda - \mu}{\sigma}\right) \tag{7}$$

Let $Z = \frac{\varepsilon_1}{\sigma}$, then $Z \sim N(0, 1)$, then:

$$P_{\text{FP}}(\text{Single}) = P\left(Z > \frac{\lambda - \mu}{\sigma}\right) = 1 - \Phi\left(\frac{\lambda - \mu}{\sigma}\right) \tag{8}$$

where $\Phi$ is the cumulative distribution function (CDF) of the standard normal distribution.

### B.1.2  PROBABILITY OF ERROR FOR AVERAGED CONFIDENCE:

For the averaged confidence $\mathcal{C}_{avg}$, the noise term is still Gaussian, $\varepsilon_{avg} \sim N(0, \sigma^2/N)$. The probability of error for averaged confidence is:

$$P_{\text{FP}}(\text{Avg}) = P(\mathcal{C}_{avg} > \lambda) = 1 - \Phi\left(\sqrt{N}\frac{(\lambda - \mu)}{\sigma}\right) \tag{9}$$

Since $\frac{(\lambda-\mu)\sqrt{N}}{\sigma} > \frac{\lambda-\mu}{\sigma}$ for $N > 1$, it follows that $P_{\text{FP}}(\text{Avg}) < P_{\text{FP}}(\text{Single})$. It indicates that simply averaging multiple observed confidence values can mitigate noise interference. Nevertheless, confidence averaging fails to address the fundamental problem, as the error rate $P_{\text{FP}}$ remains dependent on the noise standard deviation $\sigma$, increasing monotonically as $\sigma$ grows. For models with substantial intrinsic noise (large $\sigma$), the parameter inside the $\Phi$ function converges to zero, driving $P_{\text{FP}}$ toward 0.5. This indicates that high-noise models reduce to random guessing, regardless of the decision threshold $\lambda$. Therefore, the reliability of traditional approaches is severely constrained by the model's inherent noise level $\sigma$, a factor beyond our control.

### B.1.3 PROBABILITY OF ERROR FOR MAD-CALIBRATED CONFIDENCE (DEER-PRO):

$$P_{\text{FP}}(\text{calibration}) = P(\mathcal{C}_{cali} > \lambda) = P(\mathcal{C}_{avg} - \alpha \cdot \mathcal{C}_{\text{MAD}} > \lambda) \tag{10}$$

Substituting $\mathcal{C}_{avg} = \mu + \varepsilon_{avg}$, we obtain:

$$P_{\text{FP}}(\text{calibration}) = P(\mu + \varepsilon_{avg} - \alpha \cdot \mathcal{C}_{\text{MAD}} > \lambda) \tag{11}$$

Rearranging the terms in the equation yields:

$$P_{\text{FP}}(\text{calibration}) = P(\varepsilon_{avg} > \alpha \cdot \mathcal{C}_{\text{MAD}} + \lambda - \mu) \tag{12}$$

Next, we will discuss the robustness of DEER-PRo under two distinct scenarios.

**Scenario 1: Approximate estimation of $\mathcal{C}_{\text{MAD}}$**

Under our assumption where $\varepsilon_i \sim N(0, \sigma^2)$, we have $\varepsilon_{avg} \sim N(0, \sigma^2/N)$ and $E[\mathcal{C}_{\text{MAD}}] \approx 0.8\sigma$ (we will provide the proof in Section B.3). For large $N$, the law of large numbers allows us to approximate $\mathcal{C}_{\text{MAD}}$ as $0.8\sigma$. Given this assumption, we have:

$$P_{\text{FP}}(\text{calibration}) = P(\mathcal{C}_{avg} > \lambda + 0.8\sigma\alpha) \tag{13}$$

The above equation reveals that DEER-PRo fundamentally differs by employing an adaptive threshold that scales with noise:

$$\lambda_{\text{effective}} = \lambda + 0.8\alpha\sigma \tag{14}$$

We proceed to reformulate the equation by transforming $P_{\text{FP}}(\text{calibration})$ into the cumulative distribution function (CDF) of the standard normal distribution $\Phi$. For Equation (12), we substitute $\mathcal{C}_{\text{MAD}} = 0.8\sigma$ and obtain:

$$P_{\text{FP}}(\text{calibration}) = P(\varepsilon_{avg} > 0.8\sigma\alpha + \lambda - \mu) \tag{15}$$

Since $\varepsilon_{avg} \sim N(0, \sigma^2/N)$:

$$P_{\text{FP}}(\text{calibration}) = 1 - \Phi\left(\frac{(\lambda - \mu + 0.8\alpha\sigma)\sqrt{N}}{\sigma}\right) \tag{16}$$

Simplifying:

$$P_{\text{FP}}(\text{calibration}) = 1 - \Phi\left(\sqrt{N}\left(\frac{\lambda - \mu}{\sigma} + 0.8\alpha\right)\right) \tag{17}$$

When noise is minimal:

$$\lambda_{\text{effective}} \to \lambda \tag{18}$$

The calibrated method behaves like standard thresholding, maintaining high efficiency.

When noise dominates, $\frac{\lambda-\mu}{\sigma} \to 0$. The false positive rate becomes:

$$P_{\text{FP}} = 1 - \Phi(0.8\alpha\sqrt{N}) \tag{19}$$

which is independent of $\sigma$. It indicates that DEER-PRo effectively prevent early exit from reducing to random guessing ($P_{\text{FP}} \to 0.5$).

**Scenario 2: Exact computation of $\mathcal{C}_{\text{MAD}}$ without approximation.**

From $\mathcal{C}_{\text{MAD}} = \frac{1}{N}\sum_{i=1}^{N}|\mathcal{C}_i - \mathcal{C}_{avg}|$, we know that $\mathcal{C}_{MAD}$ is positive, therefore dividing both sides of the equation by this term, we obtain:

$$P(\varepsilon_{avg}/\mathcal{C}_{\text{MAD}} > \alpha + (\lambda - \mu)/\mathcal{C}_{\text{MAD}}) \tag{20}$$

Since $\mu < \lambda$, then $\alpha + (\lambda - \mu)/\mathcal{C}_{MAD} > \alpha$, so we can obtain:

$$P(\varepsilon_{avg}/\mathcal{C}_{\text{MAD}} > \alpha + (\lambda - \mu)/\mathcal{C}_{\text{MAD}}) < P(\varepsilon_{avg}/\mathcal{C}_{\text{MAD}} > \alpha) \tag{21}$$

Therefore, $P(\varepsilon_{avg}/\mathcal{C}_{MAD} > \alpha)$ is an upper bound for $P_{\text{FP}}(\text{calibration})$. For the ratio $\varepsilon_{avg}/\mathcal{C}_{MAD}$, $\varepsilon_{avg}$ represents the signal of the noise, while $\mathcal{C}_{MAD}$ represents the internal disorder of the noise. Therefore, we can define $\varepsilon_{avg}/\mathcal{C}_{MAD}$ as the Signal-to-Noise Ratio (SNR).

Next, let us analyze the properties of SNR. Under our assumption where $\varepsilon_i \sim N(0, \sigma^2)$, we have $\varepsilon_{avg} \sim N(0, \sigma^2/N)$ and $E[\mathcal{C}_{\text{MAD}}] \approx 0.8\sigma$. Since both $\varepsilon_{avg}$ and $\mathcal{C}_{MAD}$ are proportional to $\sigma$, we can write SNR as:

$$\text{SNR} = (\sigma * Z_{avg})/(\sigma * Z_{\text{MAD}}) = Z_{avg}/Z_{\text{MAD}} \tag{22}$$

where $Z_{avg} \sim N(0, 1/N)$ is a standardized noise mean, and $Z_{mad}$ is a random variable related to $MAD/\sigma$ whose distribution does not depend on $\sigma$. Hence, the probability distribution of the SNR is independent of the model noise standard deviation $\sigma$.

$$P_{\text{FP}}(\text{calibration}) < P(Z_{avg}/Z_{\text{MAD}} > \alpha) \tag{23}$$

Therefore, $P_{\text{FP}}(\text{calibration})$ is influenced only by the number of prompts $N$ and the signal-to-noise ratio threshold $\alpha$, where larger values of $N$ and $\alpha$ lead to lower error rates.

Through the transformation from an absolute threshold test to a self-normalized SNR test, our MAD strategy effectively decouples decision-making from the model's intrinsic and uncontrollable noise level $\sigma$. In contrast to traditional approaches that break down under high-noise conditions, our method delivers consistent, robust performance independent of model noise levels.

## B.2 ANALYSIS OF MAD-CALIBRATED STRATEGY'S SUPERIOR PERFORMANCE

In this section, we formally prove based on the event space that the false positive probability of the MAD strategy, $P_{\text{FP}}(\text{MAD})$ ($P_{\text{FP}}(\text{calibration})$), is significantly superior to that of the simple averaging strategy, $P_{\text{FP}}(\text{Avg})$, and consequently also outperforms $P_{\text{FP}}(\text{Single})$.

**Theorem.** The false positive probability of the MAD-calibrated strategy satisfies $P_{\text{FP}}(\text{MAD}) \leq \rho \cdot P_{\text{FP}}(\text{Avg})$, where $\rho = O(\exp(-\Theta(N)))$ is an exponentially decaying factor in the number of prompts $N$.

**Proof.** The false positive events are defined as:

$$E_{\text{Avg}} = \{\varepsilon : \varepsilon_{\text{avg}} > c\} \tag{24}$$

$$E_{\text{MAD}} = \{\varepsilon : \varepsilon_{\text{avg}} - \alpha \cdot \text{MAD} > c\} \tag{25}$$

where $c = \lambda - \mu > 0$ is the threshold gap, $\text{MAD} = \frac{1}{N}\sum_{i=1}^{N}|\varepsilon_i - \varepsilon_{\text{avg}}|$ is the mean absolute deviation, $\varepsilon_{\text{avg}} = \frac{1}{N}\sum_{i=1}^{N}\varepsilon_i$ is the average noise, and there are $N$ independent noise terms $\varepsilon_i \sim N(0, \sigma^2)$ for $i = 1, \ldots, N$.

Since $\alpha > 0$ and $\text{MAD} \geq 0$ by definition, we have:

$$\varepsilon_{\text{avg}} - \alpha \cdot \text{MAD} \leq \varepsilon_{\text{avg}} \tag{26}$$

Therefore, if $\varepsilon_{\text{avg}} - \alpha \cdot \text{MAD} > c$, then necessarily $\varepsilon_{\text{avg}} > c$. This establishes:

$$E_{\text{MAD}} \subseteq E_{\text{Avg}} \tag{27}$$

Consequently:

$$P_{\text{FP}}(\text{MAD}) = P(E_{\text{MAD}}) \leq P(E_{\text{Avg}}) = P_{\text{FP}}(\text{Avg}) \tag{28}$$

To quantify the improvement beyond this basic inequality, we decompose the probability using conditional probability:

$$P_{\text{FP}}(\text{MAD}) = P(E_{\text{MAD}} \cap E_{\text{Avg}}) = P(E_{\text{MAD}}|E_{\text{Avg}}) \cdot P(E_{\text{Avg}}) \tag{29}$$

Since $E_{\text{MAD}} \subseteq E_{\text{Avg}}$, we have:

$$P(E_{\text{MAD}}|E_{\text{Avg}}) = P(\varepsilon_{\text{avg}} - \alpha \cdot \text{MAD} > c|\varepsilon_{\text{avg}} > c) \tag{30}$$

This can be rewritten as:

$$P(E_{\text{MAD}}|E_{\text{Avg}}) = P\left(\text{MAD} < \frac{\varepsilon_{\text{avg}} - c}{\alpha}\middle|\varepsilon_{\text{avg}} > c\right) \tag{31}$$

Define the improvement factor:

$$\rho = P\left(\text{MAD} < \frac{\varepsilon_{\text{avg}} - c}{\alpha}\middle|\varepsilon_{\text{avg}} > c\right) \tag{32}$$

Then:

$$P_{\text{FP}}(\text{MAD}) = \rho \cdot P_{\text{FP}}(\text{Avg}) \tag{33}$$

To evaluate $\rho$, we analyze the structure of noise vectors that satisfy $\varepsilon_{\text{avg}} > c$. There are two primary patterns:

**Pattern A (Coherent Pattern):** All noise terms are close to $c$. Formally, for some small $\delta > 0$:

$$\text{Pattern A} = \{\varepsilon : |\varepsilon_i - c| < \delta \text{ for all } i = 1, \dots, N\} \tag{34}$$

Under Pattern A:

- $\varepsilon_{\text{avg}} \approx c + O(\delta/\sqrt{N})$ (by the central limit theorem)
- $\text{MAD} \leq 2\delta$ (since all values are within $2\delta$ of each other)

**Pattern B (Outlier Pattern):** A few large outliers with remaining values near zero. For example:

- $k$ values with $\varepsilon_i \approx Nc/k$ (large outliers)
- $N - k$ values with $\varepsilon_i \approx 0$

Under Pattern B:

- $\varepsilon_{\text{avg}} \approx c$
- $\text{MAD} \approx c(1 - 1/N)$ (large due to outliers)

**Probability of Pattern A:**

For a single noise term to fall in $(c - \delta, c + \delta)$:

$$P(|\varepsilon_i - c| < \delta) = \Phi\left(\frac{c + \delta}{\sigma}\right) - \Phi\left(\frac{c - \delta}{\sigma}\right) \tag{35}$$

Using Taylor expansion for small $\delta$:

$$P(|\varepsilon_i - c| < \delta) \approx \phi\left(\frac{c}{\sigma}\right) \cdot \frac{2\delta}{\sigma} = \frac{2\delta}{\sigma\sqrt{2\pi}} \exp\left(-\frac{c^2}{2\sigma^2}\right) \tag{36}$$

For all $N$ noise terms to satisfy this condition independently:

$$P(\text{Pattern A}) = \left[\frac{2\delta}{\sigma\sqrt{2\pi}} \exp\left(-\frac{c^2}{2\sigma^2}\right)\right]^N \tag{37}$$

**Probability of $\varepsilon_{\text{avg}} > c$:**

Since $\varepsilon_{\text{avg}} \sim N(0, \sigma^2/N)$:

$$P(\varepsilon_{\text{avg}} > c) = 1 - \Phi\left(\frac{c\sqrt{N}}{\sigma}\right) \tag{38}$$

Using Mill's ratio approximation for large arguments:

$$P(\varepsilon_{\text{avg}} > c) \approx \frac{\sigma}{c\sqrt{2\pi N}} \exp\left(-\frac{c^2 N}{2\sigma^2}\right) \tag{39}$$

The key insight is that Pattern A is the only pattern where MAD remains small enough to satisfy $\text{MAD} < (\varepsilon_{\text{avg}} - c)/\alpha$.

For Pattern B and other outlier-driven patterns, $\text{MAD} = O(c)$, while $\varepsilon_{\text{avg}} - c = O(1/\sqrt{N})$ when conditioned on $\varepsilon_{\text{avg}} \approx c$. Thus:

$$\text{MAD} \gg \frac{\varepsilon_{\text{avg}} - c}{\alpha} \tag{40}$$

Therefore, the improvement factor is dominated by Pattern A:

$$\rho \lesssim \frac{P(\text{Pattern A})}{P(\varepsilon_{\text{avg}} > c)} \tag{41}$$

Substituting the expressions:

$$\rho \lesssim \frac{\left[\frac{2\delta}{\sigma\sqrt{2\pi}} \exp\left(-\frac{c^2}{2\sigma^2}\right)\right]^N}{\frac{\sigma}{c\sqrt{2\pi N}} \exp\left(-\frac{c^2 N}{2\sigma^2}\right)} \tag{42}$$

Simplifying:

$$\rho \lesssim \frac{c\sqrt{N}}{\sigma} \left(\frac{2\delta}{\sigma\sqrt{2\pi}}\right)^N \exp\left(-\frac{c^2 N}{2\sigma^2} + \frac{c^2 N}{2\sigma^2}\right) \tag{43}$$

$$\rho \lesssim c\sqrt{N} \left(\frac{2\delta}{\sigma^2\sqrt{2\pi}}\right)^N \tag{44}$$

For $\delta = O(\sigma)$, let $\delta = k\sigma$ where $k$ is a constant. Then:

$$\rho \lesssim c\sqrt{N} \left(\frac{2k}{\sqrt{2\pi}}\right)^N \tag{45}$$

When $k < \sqrt{\pi/2}$, the term $(2k/\sqrt{2\pi})^N$ decays exponentially. The polynomial factor $\sqrt{N}$ is dominated by the exponential decay, yielding:

$$\rho = O(\sqrt{N} \cdot \exp(-\beta N)) = O(\exp(-\Theta(N))) \tag{46}$$

for some positive constant $\beta$.

We have established that:

$$P_{\text{FP}}(\text{MAD}) = \rho \cdot P_{\text{FP}}(\text{Avg}) \tag{47}$$

where $\rho = O(\exp(-\Theta(N)))$ decays exponentially with the number of prompts $N$.

This demonstrates that the MAD-calibrated strategy provides an exponential improvement over the simple averaging approach. $\square$

Conclusion: The MAD penalty term effectively **filters out the more probable outlier patterns while only allowing the exponentially rare coherent patterns to trigger false positives**, thus achieving superior robustness against prompt-induced noise.

### B.3 PROOF OF THE EXPECTED VALUE OF MAD

#### B.3.1 THEOREM 1

For a random variable $X \sim \mathcal{N}(\mu, \sigma^2)$, the expected value of the $\mathcal{C}_{\text{MAD}}$ is:

$$\mathbb{E}[\mathcal{C}_{\text{MAD}}] = \sigma\sqrt{\frac{2}{\pi}} \approx 0.8\sigma \tag{48}$$

### B.3.2 PROOF

**Definition.** The Mean Absolute Deviation of a random variable $X$ with mean $\mu$ is defined as:

$$\mathcal{C}_{\text{MAD}} = \mathbb{E}[|X - \mu|] \tag{49}$$

Let $X \sim \mathcal{N}(\mu, \sigma^2)$. Consider the standardized random variable:

$$Z = \frac{X - \mu}{\sigma} \sim \mathcal{N}(0, 1) \tag{50}$$

Therefore:

$$|X - \mu| = \sigma |Z| \tag{51}$$

Taking expectations on both sides:

$$\mathbb{E}[|X - \mu|] = \sigma \cdot \mathbb{E}[|Z|] \tag{52}$$

For $Z \sim \mathcal{N}(0, 1)$ with probability density function $\phi(z) = \frac{1}{\sqrt{2\pi}} e^{-z^2/2}$:

$$\mathbb{E}[|Z|] = \int_{-\infty}^{\infty} |z| \cdot \phi(z) \, dz \tag{53}$$

Due to the symmetry of the standard normal distribution about zero and the even nature of $|z|$:

$$\mathbb{E}[|Z|] = 2 \int_{0}^{\infty} z \cdot \phi(z) \, dz = \frac{2}{\sqrt{2\pi}} \int_{0}^{\infty} z \cdot e^{-z^2/2} \, dz \tag{54}$$

Let $u = z^2/2$, then $du = z \, dz$. When $z = 0$, $u = 0$; when $z \to \infty$, $u \to \infty$.

$$\int_{0}^{\infty} z \cdot e^{-z^2/2} \, dz = \int_{0}^{\infty} e^{-u} \, du = \left[ -e^{-u} \right]_{0}^{\infty} = 1 \tag{55}$$

Substituting back:

$$\mathbb{E}[|Z|] = \frac{2}{\sqrt{2\pi}} \cdot 1 = \sqrt{\frac{2}{\pi}} \tag{56}$$

Therefore:

$$\mathcal{C}_{\text{MAD}} = \mathbb{E}[|X - \mu|] = \sigma \cdot \mathbb{E}[|Z|] = \sigma \sqrt{\frac{2}{\pi}} \approx 0.8\sigma \tag{57}$$

## C  MORE EXPERIMENT SETUP

**Metrics.** The goal of DEER is to maintain the correctness performance of LRMs while avoiding the redundant token overhead caused by overthinking. To this end, we selected *Accuracy* (ACC) and *Generation Length* (LEN) as the evaluation metrics. *Accuracy* (ACC) is calculated as follows: $Accuracy = \frac{1}{N} \sum_{i=1}^{N} \mathbb{I}\{\mathcal{M}(\mathcal{LRM}(x_i)) = y_i\}$, where $x_i$ is the question and $y_i$ is the ground-truth answer from the dataset. $\mathcal{M}(\cdot)$ extracts the answer from the LRM's response. $\mathbb{I}\{\cdot\}$ is an indicator function that determines whether the inside given condition is valid. The accuracy evaluation is based on the evaluation framework publicly released by Ye et al. (2025) (LIMO). Intuitively, the longer the generated text, the greater the inference cost for LRMs. Therefore, we calculate the average generation tokens per sample to evaluate the cost as follows: *Generation Length*$(LEN) = \frac{1}{N} \sum_{i=1}^{N} |\mathcal{LRM}(x_i)|$, where $|\cdot|$ measures the number of generated tokens. For the two programming benchmarks, we use the Pass@1 metric to measure generated code correctness.

**Implementation details.** All evaluations are conducted in a Zero-shot Chain-of-Thought (CoT) setting with the following prompt: *"Please reason step by step, and put your final answer within* $\backslash boxed\{\}$*."* For the decoding strategy, we employ greedy decoding with a single sample for the correctness evaluation. The ground-truth answers to the evaluation problems in our experiments are all well-structured numerical values or options. Therefore, we apply rule-based evaluations directly to verify mathematical equivalence. We set the maximum generation length at 16,384 to ensure that the evaluation captures complete problem-solving attempts. For DEER, the answer-inducing prompt employed is: *'$\backslash n\backslash n$ Final Answer$\backslash n\backslash boxed$'* For DEER-Pro, we additionally incorporated the following three prompts: *'$\backslash n\backslash n$ Final Answer$\backslash n\backslash n$ Based on the analysis above, the answer is* $\backslash boxed$*', '$\backslash n\backslash n$ Final Answer$\backslash n\backslash n$ The correct final answer is $\backslash boxed$', '$\backslash n\backslash n$ Based on the previous thinking, I believe I already know the answer.$\backslash n$ Final Answer$\backslash n$ $\backslash boxed$'.*

---

**Algorithm 1** Dynamic Early Exit in Reasoning (DEER)

---

1: **Initialization:** Large Reasoning Language Model LRM($\cdot$), zero-shot-CoT $zs\_cot$, question, answer inducer prompt $\boldsymbol{I}$, set of action transition points $\mathbb{P}$, end-of-thinking delimiter $\langle$/think$\rangle$, maximum length $max\_len$, and confidence threshold $\lambda$.
2: $\boldsymbol{x} \leftarrow zs\_cot + $ question, $\boldsymbol{r} \leftarrow []$
3: **while** $len(\boldsymbol{x}) < max\_len$ **do**
4:      $y \leftarrow \text{LRM}(\boldsymbol{x})$
5:      **if** $y \in \mathbb{P}$ **then**                    ▷ Generate thoughts until meets action transition points
6:          $A \leftarrow \text{LRM}(\boldsymbol{x} + \boldsymbol{I})$             ▷ Prompt LRM to generate trial answer tokens
7:          Get $\mathcal{C}$ according to Equation 4          ▷ Calculate the confidence of the trial answer
8:          **if** $\mathcal{C} > \lambda$ **then**
9:              $\boldsymbol{x} \leftarrow \boldsymbol{x} + \langle$/think$\rangle$, $\boldsymbol{r} \leftarrow \boldsymbol{r} + \langle$/think$\rangle$          ▷ Exit when thinking is sufficient
10:          **end if**
11:      **else**
12:          $\boldsymbol{x} \leftarrow \boldsymbol{x} + y$, $\boldsymbol{r} \leftarrow \boldsymbol{r} + y$
13:      **end if**
14: **end while**
15: **return** $\boldsymbol{r}$

---

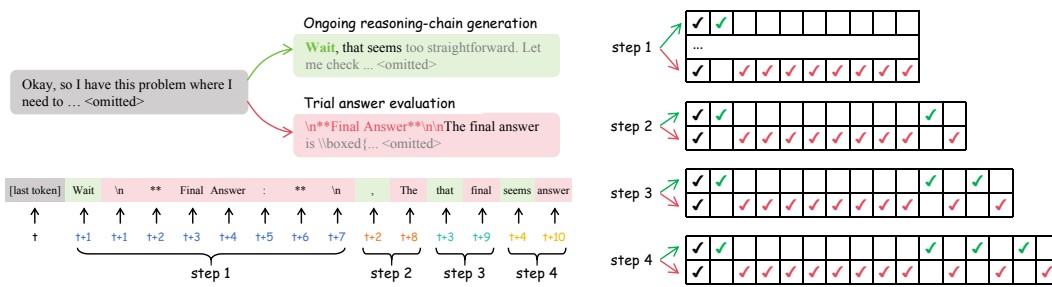

Figure 9: Branch-parallel decoding and dynamic KV cache management.

## D    MORE METHOD DETAILS

Fig. 9 illustrates the workflow of the proposed Branch-Parallel Decoding Acceleration. Algorithm 1 presents the pseudocode of DEER.

## E    MORE BENCHMARK DESCRIPTIONS.

**Benchmarks.** To thoroughly evaluate the models' performance across various reasoning capabilities, we have chosen 6 math reasoning benchmarks, 1 science benchmarks, and 3 coding benchmarks as follows:

MATH BENCHMARKS:

- **GSM8K** is a well-curated collection of 1,319 problems in elementary mathematics. This benchmark is specifically designed to evaluate multi-step reasoning in foundational math tasks. Problems typically involve two to eight sequential operations, relying primarily on basic arithmetic performed over multiple intermediate steps.

- **MATH-500** is a challenging benchmark comprising competition-level problems drawn from diverse high school mathematics domains, including Prealgebra, Algebra, and Number Theory. For consistency with previous research, we adopt the same 500-problem subset originally curated by OpenAI for evaluation.

- **AMC 2023** contains 40 mathematical problems, covering algebra, geometry, number theory, and combinatorics. The American Mathematics Competitions (AMC), organized by the Mathematical Association of America (MAA), are prestigious contests designed to develop problem-solving skills and identify mathematical talent. For evaluation, we used 40 questions from AMC 23 in LIMO.

- **AIME 2024** comprises 30 challenge problems selected from the 2024 American Invitational Mathematics Examination (AIME). This prestigious contest evaluates participants' mathematical reasoning abilities across diverse domains, including arithmetic, algebra, counting, geometry, number theory, probability, and other secondary school math topics. A distinctive feature of the AIME is its answer format: all solutions must be integers between 000 and 999 (inclusive). Each problem is categorized by difficulty level (1–5) according to the Art of Problem Solving (AoPS) scale. Beyond these three math problems, we also conducted evaluations on scientific questions.

- **AIME 2025** comprises 30 challenge problems selected from the 2025 American Invitational Mathematics Examination (AIME).

- **OlympiadBench** OlympiadBench is an Olympiad-level bilingual multimodal scientific benchmark dataset that aims to challenge and evaluate the advanced capabilities of Large Language Models and Large Multimodal Models. It features 8,476 problems sourced from mathematics and physics competitions at the Olympiad level, including those from the Chinese college entrance exam. Our experimental evaluation selects the same subset of 675 samples as used in LIMO, allowing for direct rule-based evaluation of the generated answers.

**SCIENCE BENCHMARKS**:

- **GPQA** is a PhD-level benchmark consisting of high-quality questions spanning physics, chemistry, and biology subdomains. Notably, domain experts with PhDs in these fields achieved only 69.7% accuracy on this dataset. For our experiments, we specifically select the highest quality subset, known as **GPQA Diamond** (composed of 198 questions).

**PROGRAMMING BENCHMARKS**:

- **HumanEval** is proposed by OpenAI, containing 164 hand-crafted (to avoid data leakage) Python programming tasks focusing on basic algorithms, each with function signatures, docstrings, canonical solutions, and unit tests.

- **BigCodeBench** is designed as a real-world-oriented benchmark, which includes 1,140 tasks requiring interactions with 139 libraries and diverse function calls.

- **LiveCodeBench** is a newly proposed benchmark dataset designed to evaluate the capabilities of large language models in code generation and related tasks. It aims to mitigate issues such as test set contamination found in existing benchmarks by emphasizing scenarios beyond code generation, ensuring high-quality problem sources, adequate test cases, and balanced difficulty levels. The dataset comprises problems sourced from well-known competitive programming platforms like AtCoder, LeetCode, and CodeForces, collected from specific time windows. Our evaluation is based on **LiveCodeBench-v5**, which contains 880 programming problems collected from May 2023 to January 2025.

## F  COMPUTATION SOURCE

In our experiments, $8 \times 80$g memory H100 was used to perform evaluations.

## G  MORE LRM DESCRIPTIONS.

In this work, we validate the effectiveness of DEER across 12 reasoning models. The evaluated models include: Qwen3-1.7B, Qwen3-4B, Qwen3-8B, Qwen3-14B, Qwen3-32B, DeepSeek-R1-Distill-Qwen-1.5B, DeepSeek-R1-Distill-Qwen-7B, DeepSeek-R1-Distill-Qwen-14B, DeepSeek-R1-Distill-Qwen-32B, QwQ-32B, DeepSeek-R1-671B, and Llama-3.1-Nemotron-Nano-8B-v1. All models in the DeepSeek-R1-Distill-Series were supervised fine-tuned using reasoning data generated by the DeepSeek-R1 model. The Qwen3-1.7B, Qwen3-4B, Qwen3-8B, and Qwen3-14B models were trained using a method known as Strong-to-Weak Distillation. Trained via reinforcement learning, the non-distilled models QwQ-32B and Qwen3-32B demonstrate competitive performance on reasoning benchmarks, matching that of DeepSeek-R1-671B. Due to computational constraints, we implemented a quantized version of Deepseek-R1 based on KTransformers (kvcache ai, 2025).

## H    COMPUTATIONAL COST ANALYSIS

In this section, we provide a theoretical analysis to demonstrate that DEER effectively reduces computational costs. Let $L$ denote the total length generated by the original CoT method, and $\alpha$ represent DEER's compression ratio relative to $L$, such that DEER generates a sequence of length $\alpha L$. Then, we define $k$ as the number of answer induction triggers within these $\alpha L$ tokens of reasoning and $m$ as the average length generated per answer induction, which is typically a small constant. During transformer inference, the primary computational overhead stems from attention calculations, which constitutes our main focus. Assuming the generation process employs key-value caching technology, each new token only needs to compute attention with the cached key-value pairs.

### H.1    COMPUTATIONAL COST ANALYSIS ON TIME

For the original CoT method, the computational cost is:

$$T = O(1) + O(2) + \cdots + O(L) = O(L^2) \tag{58}$$

For our DEER, The computational cost comprises two components: $\alpha L$ forward passes in the main reasoning chain and $km$ forward passes for answer inducing.

First, we calculate the cost of the main reasoning chain:

$$T_{\text{main}} = \sum_{t=1}^{\alpha L} t = \frac{\alpha L(\alpha L + 1)}{2} = O\left(\alpha^2 L^2\right) \tag{59}$$

Next, for the computational overhead during answer inducing, we first calculate the time cost of a single inducing. Suppose the $j$-th answer inducing is triggered at position $p_j$, yielding a cost of:

$$C_{\text{single}} = \sum_{i=1}^{m}(p_j + i - 1) = m \cdot p_j + \sum_{i=1}^{m}(i - 1) = m \cdot p_j + \frac{m(m-1)}{2} \tag{60}$$

Assuming the inducing positions $p_j$ are uniformly distributed over the interval $[0, \alpha L]$, the average inducing position is $E[p_j] \approx \frac{\alpha L}{2}$. Hence, the total cost of answer inducing is:

$$C_{\text{induce}} \approx k \cdot m \cdot \frac{\alpha L}{2} + k \cdot \frac{m(m-1)}{2} = O(k \cdot m \cdot \alpha L) + O(k \cdot m^2) \tag{61}$$

Even in the worst-case scenario where most trigger points $p_j$ cluster near the end of reasoning, the average inducing position is $E[p_j] \approx \alpha L$. The total cost of answer inducing is:

$$C_{\text{induce}} \approx k \cdot m \cdot \alpha L + k \cdot \frac{m(m-1)}{2} = O(k \cdot m \cdot \alpha L) + O(k \cdot m^2) \tag{62}$$

As the length of each answer inducing $m$ is negligible compared to the reasoning length $L$, we have:

$$C_{\text{induce}} \approx O(k \cdot m \cdot \alpha L) \tag{63}$$

Finally, the total cost of DEER is:

$$C_{\text{DEER}} = C_{\text{main}} + C_{\text{induce}} = O(\alpha^2 L^2) + O(k \cdot m \cdot \alpha L) \tag{64}$$

DEER reduces the quadratic term from $O(L^2)$ to $O(\alpha^2 L^2)$ while only introducing a linear term $O(k \cdot m \cdot \alpha L)$. Since $k, m \ll L$ in long chain-of-thought reasoning, the savings from the quadratic term reduction far exceed the overhead of the additional linear term. This analysis effectively explains the superlinear speedup phenomenon observed in Section 4.4.

### H.2    COMPUTATIONAL COST ANALYSIS ON MEMORY

The memory overhead analysis can be decomposed into two components: primary memory consumption from the KV cache and additional overhead from parallel decoding operations.

**Peak Memory Reduction.** The dominant memory overhead in modern LLM inference stems from the storage of attention keys and values in the KV cache, whose size scales linearly with the processed sequence length. Standard Chain-of-Thought (CoT) approaches necessitate maintaining KV cache for all $L$ tokens, resulting in memory complexity of $O(L)$. Through early termination at position $\alpha L$ where $\alpha < 1$, DEER effectively reduces the peak sequence length during inference from $L$ to $\alpha L$. Consequently, the peak KV cache memory consumption is reduced from $O(L)$ to $O(\alpha L)$.

This memory reduction proves particularly valuable when processing long-context reasoning tasks. Beyond reducing peak memory requirements for individual requests, this approach enables systems to accommodate increased concurrent requests under identical memory constraints during batch processing, thereby enhancing overall throughput.

**Additional Overhead for Parallel Decoding.** The proposed parallel decoding variant, which performs answer induction forward passes concurrently with main reasoning, introduces minimal additional memory overhead. This efficiency is achieved through prefix caching and sharing mechanisms implemented in modern inference frameworks such as vLLM. When multiple reasoning branches share a common prefix sequence, the corresponding portions of their KV caches require only single storage in physical memory through technologies such as vLLM's PagedAttention.

During parallel decoding, the answer induction branch incurs virtually no additional KV cache overhead, as it fully leverages the KV cache already computed by the main reasoning branch. The only marginal additional memory requirement arises from storing a limited number of tokens representing the answer induction branch's decoding state.

For code generation tasks, our implementation incorporates specific optimizations whereby only the initial 50 tokens are generated for confidence estimation. This design choice represents an implementation detail rather than a core methodological contribution. Experimental validation confirms that utilizing partial answer tokens for early-exit confidence calculation remains effective for coding tasks.

# I INVESTIGATION OF REASONING TRANSITION MONITORS

In Section 4.3 of the main text, our experiments reveal that the choice of Reasoning Transition Monitor exerts subtle effects on DEER, primarily manifested in how the number of potential early-exit opportunities affects the final generation length. In this section, we investigate the underlying connections between linguistic marker-based and entropy-based monitoring approaches.

Table 10 presents a comparative analysis of average token entropy between linguistic markers and other tokens across multiple datasets and models. Our findings reveal that linguistic markers exhibit significantly higher entropy compared to other tokens, suggesting that the linguistic marker-based approach inherently targets high-entropy positions where multiple candidate actions exist.

Additionally, we compute cosine similarity scores between consecutive tokens in the final layer's hidden state representations, comparing linguistic markers with their adjacent tokens against regular token pairs. The similarity metric serves as an indicator of the model's reasoning coherence: high similarity reflects continuous, coherent reasoning processes, whereas low similarity signals the occurrence of reasoning transitions. The results presented in Table 11 demonstrate that linguistic markers exhibit substantially lower similarity scores, indicating disruptions in representational continuity.

Collectively, these experiments provide compelling evidence that large reasoning language models do not undergo uncertain states silently; instead, they explicitly express uncertainty through language. The external linguistic markers leveraged by DEER constitute direct manifestations of internal state transitions, thereby providing strong empirical support for the theoretical foundations of our approach.

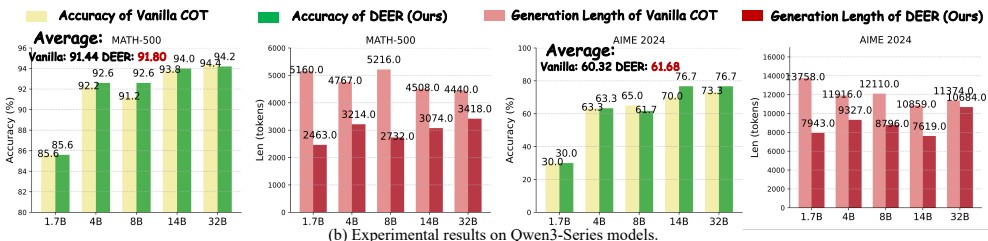

Figure 10: Experimental results of DEER compared to Vanilla CoT across Qwen3-Series models of varying sizes on MATH-500 and AIME 2024.

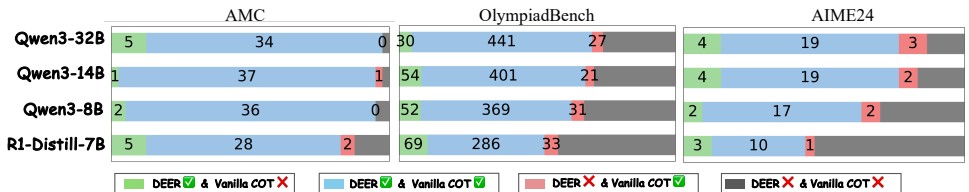

Figure 11: More detailed experimental results of DEER compared to Vanilla CoT. $\sqrt{}$ denotes a correct answer, and $\times$ denotes an incorrect answer.

## J  INVESTIGATION INTO THE REASONS BEHIND DEER'S THRESHOLD ROBUSTNESS

In Section 4.3 of the main text, we demonstrate DEER's robustness to the threshold $\lambda$ through experiments across various models and datasets. In this section, we investigate the underlying source of this robustness. We analyze the confidence scores of induced answers at all potential exit positions across three models on three mathematical reasoning datasets, calculating the proportion of scores falling within three distinct intervals. Specifically, 0–0.9 represents the low-confidence interval, 0.97–1.0 represents the high-confidence interval, and 0.9–0.97 constitutes the error-prone gray zone.

The results presented in Table 12 reveal that the model's confidence distribution exhibits a pronounced polarization phenomenon. The vast majority of cases concentrate at either the highly confident or insufficiently confident extremes, with minimal presence in the intermediate range (the error-prone gray zone). When our method induces the model to generate final answers, the confidence scores follow a distinctive U-shaped distribution, with remarkably low probability mass between 0.9 and 0.97. This phenomenon indicates that when the model possesses sufficient certainty about an answer based on its preceding reasoning chain, it generates the answer with exceptionally high probability (typically exceeding 0.99). Conversely, when uncertainty exists, the assigned probability drops substantially.

Furthermore, we observe that all three models exhibit higher proportions of high-confidence scores compared to low-confidence scores on simpler problems (GSM8K); While on more challenging problems (AIME24), the proportion of low-confidence scores exceeds that of high-confidence scores. This observation further validates the rationality of the DEER method: the confidence assigned to trial answers accurately reflects whether the existing reasoning is sufficient to solve the problem. Consequently, confidence scores are generally lower on difficult problems, leading to many failed early-exit attempts in the initial stages. This pattern also explains DEER's varying performance across different problem difficulties. On simpler problems, the model demonstrates sufficient confidence, resulting in better compression effects. On challenging problems, the model becomes more cautious, yielding weaker compression but maintaining satisfactory accuracy. This adaptive behavior shows that DEER naturally balances computational efficiency and solution quality based on problem complexity.

Table 2: Comparison of **Vanilla**, **DEER**, and **DEER-PRo** across multiple models and datasets. Acc = accuracy (%), Len = average tokens, CR = compression ratio.

| Method | GSM8K | | | MATH | | | AMC | | | AIME | | | GPQA | | | Overall | |
|---|---|---|---|---|---|---|---|---|---|---|---|---|---|---|---|---|---|
| | Acc | Len | CR | Acc | Len | CR | Acc | Len | CR | Acc | Len | CR | Acc | Len | CR | Acc | CR |
| *DeepSeek-R1-Distill-Qwen-1.5B* | | | | | | | | | | | | | | | | | |
| Vanilla | 76.1 | 1,617 | 100% | 69.0 | 6,018 | 100% | 52.5 | 8,819 | 100% | 23.3 | 13,702 | 100% | 7.1 | 13,029 | 100% | 45.6 | 100% |
| DEER | 74.7 | 984 | 60.9% | 67.8 | 2,497 | 41.5% | 60.0 | 5,496 | 62.3% | 23.3 | 9,557 | 69.7% | 12.1 | 5,762 | 44.2% | 47.6 | **55.7%** |
| DEER-PRo | 77.3 | 1,062 | 65.7% | 70.0 | 2,891 | 48.0% | 62.5 | 5,701 | 64.6% | 26.7 | 10,390 | 75.8% | 14.5 | 6,820 | 52.3% | **50.2** | 61.3% |
| *Qwen3-4B* | | | | | | | | | | | | | | | | | |
| Vanilla | 94.1 | 2,175 | 100% | 92.2 | 4,767 | 100% | 87.5 | 7,443 | 100% | 63.3 | 11,916 | 100% | 46.5 | 9,294 | 100% | 76.7 | 100% |
| DEER | 94.5 | 1,250 | 57.5% | 92.6 | 3,214 | 67.4% | 87.5 | 4,906 | 65.9% | 63.3 | 9,327 | 78.3% | 47.5 | 3,275 | 35.2% | 77.1 | **60.9%** |
| DEER-PRo | 94.5 | 1,301 | 59.8% | 93.0 | 3,517 | 73.8% | 92.5 | 5,153 | 69.2% | 65.0 | 9,651 | 81.0% | 49.2 | 3,750 | 40.3% | **78.8** | 64.8% |
| *Qwen3-1.7B* | | | | | | | | | | | | | | | | | |
| Vanilla | 90.1 | 2,045 | 100% | 85.6 | 5,160 | 100% | 70.0 | 8,637 | 100% | 30.0 | 13,758 | 100% | 35.9 | 9,271 | 100% | 62.3 | 100% |
| DEER | 90.3 | 1,066 | 52.1% | 85.6 | 2,463 | 47.7% | 70.0 | 4,673 | 54.1% | 30.0 | 7,943 | 57.7% | 43.4 | 3,549 | 38.3% | 63.9 | **50.0%** |
| DEER-PRo | 90.7 | 1,261 | 61.7% | 87.2 | 2,702 | 52.4% | 75.0 | 5,143 | 59.5% | 35.0 | 8,644 | 62.8% | 44.5 | 3,960 | 42.7% | **66.5** | 55.8% |

Table 3: Experimental results on programming tasks. Acc = accuracy (%), Tok. = average tokens, CR = compression ratio.

| Model | Method | HumanEval | | | BigCodeBench | | | LiveCodeBench | | | Overall | |
|---|---|---|---|---|---|---|---|---|---|---|---|---|
| | | Acc | Tok. | CR | Acc | Tok. | CR | Acc | Tok. | CR | Acc | CR |
| **R1-Distill-Qwen Series** | | | | | | | | | | | | |
| 32B | Vanilla | 91.5 | 3,861 | 100% | 44.5 | 5,459 | 100% | 56.0 | 9,109 | 100% | 64.0 | 100% |
| | DEER | 93.9 | 1,254 | 32.5% | 46.1 | 1,929 | 35.3% | 56.6 | 3,677 | 40.4% | **65.5** | **36.1%** |
| 14B | Vanilla | 89.0 | 4,039 | 100% | 40.9 | 4,806 | 100% | 52.7 | 9,259 | 100% | 60.9 | 100% |
| | DEER | 90.9 | 1,000 | 24.8% | 40.7 | 1,583 | 32.9% | 52.1 | 4,000 | 43.2% | **61.2** | **33.6%** |
| 7B | Vanilla | 78.6 | 5,666 | 100% | 26.1 | 8,516 | 100% | 38.4 | 10,482 | 100% | 47.7 | 100% |
| | DEER | 78.6 | 913 | 16.1% | 25.2 | 1,605 | 18.8% | 40.3 | 2,582 | 24.6% | **48.0** | **19.9%** |
| **Qwen3 Series** | | | | | | | | | | | | |
| 14B | Vanilla | 93.3 | 3,277 | 100% | 44.6 | 5,072 | 100% | 73.4 | 8,203 | 100% | 70.4 | 100% |
| | DEER | 93.9 | 1,118 | 34.1% | 44.3 | 792 | 15.6% | 74.1 | 5,437 | 66.3% | **70.8** | **38.7%** |
| 8B | Vanilla | 85.4 | 3,904 | 100% | 37.9 | 6,994 | 100% | 64.7 | 8,871 | 100% | 62.7 | 100% |
| | DEER | 87.8 | 793 | 20.3% | 41.1 | 608 | 8.7% | 65.1 | 4,257 | 48.0% | **64.7** | **25.7%** |
| 4B | Vanilla | 91.5 | 3,768 | 100% | 36.1 | 7,804 | 100% | 64.7 | 8,789 | 100% | 64.1 | 100% |
| | DEER | 92.7 | 1,050 | 27.9% | 37.1 | 826 | 10.6% | 63.5 | 5,626 | 64.0% | **64.4** | **34.2%** |
| 1.7B | Vanilla | 83.5 | 3,580 | 100% | 25.7 | 6,151 | 100% | 51.7 | 9,447 | 100% | 53.6 | 100% |
| | DEER | 84.1 | 2,236 | 62.5% | 28.3 | 2,104 | 34.2% | 51.4 | 8,425 | 89.2% | **54.6** | **62.0%** |

## K  MORE EXPERIMENTAL RESULTS

**More experiments across Model Sizes on Qwen3.** The performance of the Qwen3-series models across model sizes and reasoning difficulty in Fig. 10 is consistent with the findings presented in Section 4.2.

**Performance on SoTA Reasoning Models.** We evaluated DEER's effectiveness on two state-of-the-art reasoning models: Qwen3-32B (representing dense models) and Deepseek-R1 671B (representing MoE models). To fully leverage their reasoning capabilities, we set their maximum sequence lengths to the officially recommended 32k and 16k, respectively. The impact of max length will be further discussed in the next section. Due to computational constraints, we implemented a quantized version of Deepseek-R1 based on KTransformers (kvcache ai, 2025). Fig. 12 provides a close look at DEER's performance on two challenging datasets, AIME and MATH.

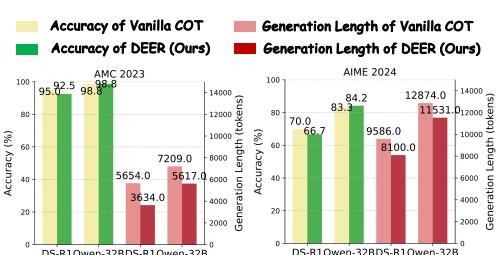

Figure 12: Performance on SoTA models.

Table 4: Additional threshold sensitivity experiments across more models and tasks.

| Qwen3-14B | GSM8K | MATH | AMC |
|---|---|---|---|
| Vanilla | 95.1 | 93.8 | 95.0 |
| 0.80 | 96.0 | 93.8 | 93.8 |
| 0.85 | 95.7 | **94.4** | 93.8 |
| 0.90 | **96.1** | 93.8 | 95.0 |
| 0.95 | 96.0 | 94.0 | **95.6** |
| 0.97 | 95.7 | 93.8 | 94.4 |
| **Qwen3-8B** | **GSM8K** | **MATH** | **AMC** |
| Vanilla | 94.9 | 91.2 | 87.5 |
| 0.80 | 95.2 | 91.4 | 88.8 |
| 0.85 | 94.9 | 92.0 | 90.0 |
| 0.90 | **95.5** | 92.8 | 91.3 |
| 0.95 | 95.3 | **93.2** | **92.5** |
| 0.97 | 95.3 | 93.0 | **92.5** |

The results show that DEER maintains competitive accuracy (with R1 making only one additional error on each dataset) while significantly reducing sequence length by 10.4% - 35.7%.

**Performance Trends across More Model Sizes and Benchmarks.** To provide a more comprehensive demonstration of DEER's effectiveness and facilitate comparison for researchers, we present experimental results on seven reasoning benchmarks and eleven large reasoning language models. Fig. 13 compares the experimental results between DEER and vanilla CoT, demonstrating that the conclusions drawn in the main text of the paper hold consistently across more benchmarks and additional models.

In addition to popular reasoning models, we also evaluate DEER on less commonly used models. As mentioned in the Limitations section, Llama-3.1-Nemotron-Nano-8B-v1 consistently exhibits low confidence in generating intermediate answers, resulting in a significantly lower early stopping rate during evaluation compared to mainstream models (Qwen3-8B: 80%, R1-Distill-Qwen-7B: 85%, Llama-3.1-Nemotron-Nano-8B-v1: 55%). Consequently, as shown in Table 5, the improvement in reasoning efficiency for Llama-3.1-Nemotron-Nano-8B-v1 is limited. Nevertheless, DEER still effectively mitigates overthinking in this model, as evidenced by its ability to prevent subsequent reasoning steps from altering correct answers into incorrect ones through early stopping.

**Performance with Different Decoding Configurations.** As the configuration of DeepSeek-R1-Distill-Series models recommends a maximum length of 16k (16,384), we evaluate Qwen3-14B under the same setting in the main experiments to maintain setup consistency. In practice, this length is sufficient for most real-world applications. In addition, to ensure experimental stability and reproducibility, we employ greedy decoding in the main experiments. Nevertheless, to provide a more comprehensive assessment of DEER's performance, we further conduct experiments on larger variants of the Qwen3-series models (8B, 14B, 32B) using the officially recommended decoding strategy (max_len = 32768, top_p = 0.95, temperature = 0.6). Fig. 8 shows that DEER remains significantly effective under these configurations, demonstrating the robustness of our approach.

**Performance with Different Multi-Token Confidence Averaging Methods.** In the main text, we mentioned adopting the geometric mean strategy for calculating multi-token answer confidence scores, as it better aligns with the multiplicative nature of joint probability computation in language models and exhibits higher sensitivity to low probability values. In this section, we supplement our analysis with comparative experiments using arithmetic mean calculation (DEER-AM), employing the same early-exit threshold of 0.95. The results in Table 7 demonstrate that DEER-AM exhibits a significant decrease in accuracy compared to DEER-GM, while achieving marginal improvements in compression ratio. This indicates that the arithmetic mean dilutes the contribution of low-valued tokens, resulting in overall inflated confidence scores and consequently leading to premature incorrect exits. Therefore, we recommend using the geometric mean for estimating true confidence scores.

Table 5: Results on **MATH-500** (DeepSeek-R1-Distill-Qwen-14B) with different reasoning transition monitors. DEER(W) denotes transition via *Wait*, DEER(A) via *Alternatively*, and DEER(Ent) via entropy threshold. Chunk Size denotes the length (token numbers) of one reasoning chunk (thought), and Chunk Num denotes the number of reasoning chunks.

| Method | Accuracy | Tokens | Chunk Size | Chunk Num | Exit Ratio | Exit Acc. |
|---|---|---|---|---|---|---|
| Vanilla | 88.6 | 3815 | – | – | – | – |
| DEER(W) | $89.6_{+1.0}$ | $\textbf{2572}_{-32.6\%}$ | 259.5 | 14.7 | 87.6% | **93.4%** |
| DEER(A) | $\textbf{90.8}_{+2.2}$ | $2775_{-27.3\%}$ | 719.8 | 5.3 | 54.8% | 91.2% |
| DEER(Ent) | $90.2_{+1.6}$ | $2339_{-38.7\%}$ | 183.7 | 20.8 | **90.2%** | 93.0% |

Table 6: Comparison of **Vanilla**, **DEER-W**, and **DEER-Ent** across multiple models and datasets. Acc = accuracy (%), Len = average tokens, CR = compression ratio.

| Method | GSM8K | | | MATH | | | AMC | | | AIME | | | GPQA | | | Overall | |
|---|---|---|---|---|---|---|---|---|---|---|---|---|---|---|---|---|---|
| | Acc | Len | CR | Acc | Len | CR | Acc | Len | CR | Acc | Len | CR | Acc | Len | CR | Acc | CR |
| *DeepSeek-R1-Distill-Qwen-7B* | | | | | | | | | | | | | | | | | |
| Vanilla | 89.6 | 1,484 | 100% | 87.4 | 3,858 | 100% | 78.8 | 6,792 | 100% | 41.7 | 13,765 | 100% | 23.7 | 10,247 | 100% | 64.2 | 100% |
| DEER-W | 90.6 | 917 | 61.8% | 89.8 | 2,143 | 55.5% | 85.0 | 4,451 | 65.5% | 49.2 | 9,839 | 71.5% | 31.3 | 5,469 | 53.4% | **69.2** | **61.5%** |
| DEER-Ent | 90.8 | 876 | 59.0% | 89.2 | 2,261 | 58.6% | 85.0 | 4,072 | 60.0% | 48.4 | 8,961 | 65.1% | 29.6 | 5,037 | 49.2% | 68.6 | 58.4% |
| *Qwen3-14B* | | | | | | | | | | | | | | | | | |
| Vanilla | 95.1 | 2,047 | 100% | 93.8 | 4,508 | 100% | 95.0 | 7,139 | 100% | 70.0 | 10,859 | 100% | 60.1 | 7,339 | 100% | 82.8 | 100% |
| DEER-W | 95.3 | 840 | 41.0% | 94.0 | 3,074 | 68.2% | 95.0 | 4,763 | 66.7% | 76.7 | 7,619 | 70.2% | 57.6 | 2,898 | 39.5% | **83.7** | **57.1%** |
| DEER-Ent | 96.1 | 803 | 39.2% | 93.8 | 2,979 | 66.1% | 93.3 | 4,903 | 68.7% | 73.3 | 7,128 | 65.6% | 58.1 | 2,818 | 38.4% | 82.9 | 55.6% |
| *Qwen3-8B* | | | | | | | | | | | | | | | | | |
| Vanilla | 94.9 | 2,245 | 100% | 91.2 | 5,216 | 100% | 87.5 | 7,986 | 100% | 65.0 | 12,110 | 100% | 51.5 | 9,145 | 100% | 78.0 | 100% |
| DEER-W | 95.2 | 1,071 | 47.7% | 92.6 | 2,732 | 52.4% | 92.5 | 4,392 | 55.0% | 61.7 | 8,796 | 72.6% | 52.5 | 3,111 | 34.0% | **78.9** | **52.3%** |
| DEER-Ent | 95.8 | 1,037 | 46.2% | 93.6 | 2,789 | 53.5% | 91.3 | 4,003 | 50.1% | 63.3 | 8,328 | 68.8% | 51.5 | 3,248 | 35.5% | **79.1** | **50.8%** |

**Error Bars with 95% Confidence Intervals.** To demonstrate the statistical significance of DEER's accuracy gains, we conducted multiple experimental runs on two models and calculated error bars with 95% confidence intervals. Specifically, we performed four independent runs on GSM8K, MATH, and GPQA benchmarks. Given the limited sample sizes of AMC23 and AIME24, we increased the number of experimental repetitions to eight for these datasets. The results presented in Table 9 confirm that the accuracy improvements achieved by our method are statistically significant.

## L CASE STUDY DETAILS

In Fig. 13, we provide examples of results on MATH-500 to visually demonstrate the effectiveness of DEER. The design of DEER ensures that it follows the same reasoning process as the vanilla CoT method before early exiting. Both methods arrive at the correct answer during the first reasoning step, as shown in the green box. The difference lies in the fact that our method exits early after evaluating the confidence of the trial answer as sufficiently high, thus producing the correct result. In contrast, the vanilla CoT method proceeds to the next reasoning action. After double-checking and switching reasoning approaches, the model becomes trapped in an endless cycle of verification due to inconsistent answers from the two approaches, ultimately failing to provide a final answer. In addition, Fig. 14, 15, 16 provides additional generated examples to more comprehensively demonstrate the effectiveness of DEER's early-exit mechanism and illustrate the underlying mechanisms of the approach.

Figure 17 shows the detailed process of DEER applied on a mathematical example. It can be observed that, at each reasoning switch point (*"Wait"* token), DEER generates a trial answer and evaluates its confidence. The change in confidence is consistent with the reliability of the current reasoning chunks and trial answers. This shows that LRMs implicitly know when to leave early, and our method is simple and effective to realize such potential of the model itself.

Table 7: Comparison of **Vanilla**, **DEER-GM** (Geometric Mean), and **DEER-AM** (Arithmetic Mean) across multiple models and datasets. Acc = accuracy (%), Len = average tokens, CR = compression ratio.

| Method | GSM8K | | | MATH | | | AMC | | | AIME | | | GPQA | | | Overall | |
|---|---|---|---|---|---|---|---|---|---|---|---|---|---|---|---|---|---|
| | Acc | Len | CR | Acc | Len | CR | Acc | Len | CR | Acc | Len | CR | Acc | Len | CR | Acc | CR |
| *DeepSeek-R1-Distill-Qwen-7B* | | | | | | | | | | | | | | | | | |
| Vanilla | 89.6 | 1,484 | 100% | 87.4 | 3,858 | 100% | 78.8 | 6,792 | 100% | 41.7 | 13,765 | 100% | 23.7 | 10,247 | 100% | 64.2 | 100% |
| DEER-GM | 90.6 | 917 | 61.8% | 89.8 | 2,143 | 55.5% | 85.0 | 4,451 | 65.5% | 49.2 | 9,839 | 71.5% | 31.3 | 5,469 | 53.4% | **69.2** | **61.5%** |
| DEER-AM | 90.2 | 832 | 56.1% | 88.2 | 1,879 | 48.7% | 80.0 | 3,872 | 57.0% | 43.3 | 8,095 | 58.8% | 22.6 | 4,116 | 40.2% | 64.9 | 52.2% |
| *Qwen3-14B* | | | | | | | | | | | | | | | | | |
| Vanilla | 95.1 | 2,047 | 100% | 93.8 | 4,508 | 100% | 95.0 | 7,139 | 100% | 70.0 | 10,859 | 100% | 60.1 | 7,339 | 100% | 82.8 | 100% |
| DEER-GM | 95.3 | 840 | 41.0% | 94.0 | 3,074 | 68.2% | 95.0 | 4,763 | 66.7% | 76.7 | 7,619 | 70.2% | 57.6 | 2,898 | 39.5% | **83.7** | **57.1%** |
| DEER-AM | 95.3 | 811 | 39.6% | 92.4 | 2,620 | 58.1% | 90.0 | 4,513 | 63.2% | 63.3 | 6,933 | 63.8% | 53.6 | 2,508 | 34.2% | 78.9 | 51.8% |
| *Qwen3-8B* | | | | | | | | | | | | | | | | | |
| Vanilla | 94.9 | 2,245 | 100% | 91.2 | 5,216 | 100% | 87.5 | 7,986 | 100% | 65.0 | 12,110 | 100% | 51.5 | 9,145 | 100% | 78.0 | 100% |
| DEER-GM | 95.2 | 1,071 | 47.7% | 92.6 | 2,732 | 52.4% | 92.5 | 4,392 | 55.0% | 61.7 | 8,796 | 72.6% | 52.5 | 3,111 | 34.0% | **78.9** | **52.3%** |
| DEER-AM | 94.9 | 972 | 43.3% | 92.0 | 2,522 | 48.4% | 87.5 | 3,899 | 48.8% | 56.7 | 7,697 | 63.6% | 49.7 | 2,950 | 32.3% | 76.2 | 47.3% |

# M  RELATED WORK DETAILS

The advent of Open-AI o1 (OpenAI, 2025) established test-time scaling (Snell et al., 2024) as a pivotal research direction in the LLM community. This approach enhances LLMs' slow thinking capabilities, enabling breakthroughs in complex problem solving. The recent open-sourcing of DeepSeek-R1 (DeepSeek-AI et al., 2025) has further intensified interest in locally deployed reasoning models. However, two critical challenges have emerged: 1) excessively long CoT generated significantly degrades inference efficiency, and 2) growing empirical evidence (Chen et al., 2025b; Team et al., 2025a) reveals their susceptibility to overthinking – a phenomenon where models continue reasoning beyond the point of optimal output. Zhang et al. (2025c) introduces a novel benchmark named S1-Bench to test the performance of LRMs on simple tasks, evaluating the overthinking issues of these LRMs. Following the taxonomy of efficient reasoning established in (Sui et al., 2025; Wang et al., 2025a), we categorize related work into three classes: post-traning based, prompt-based, and output-based efficient reasoning methods.

**Post-training based efficient reasoning** methods use supervised fine-tuning (Yu et al., 2024; Kang et al., 2025; Xia et al., 2025; Ma et al., 2025b; Munkhbat et al., 2025; Zhu et al., 2025; Liu et al., 2024; Han et al., 2024; Qiao et al., 2025; Yu et al., 2025) with variable-length CoT data or incorporate length rewards (Team et al., 2025b; Luo et al., 2025a; Aggarwal & Welleck, 2025; Arora & Zanette, 2025; Yeo et al., 2025; Shen et al., 2025b; Qu et al., 2025; Cui et al., 2025; Dai et al., 2025; Liu et al., 2025a;b; Tu et al., 2025; Wang et al., 2025c; Dumitru et al., 2025; Li et al., 2025a; Jiang et al., 2025a; Zhang et al., 2025b) in reinforcement learning to enable the model to adaptively generate chains of thought of different lengths. However, these methods often require a large amount of computational resources and face challenges in dataset construction. Recently, some work (Hao et al., 2024; Shen et al., 2025c; Cheng & Van Durme, 2024; Dang et al., 2025; Shen et al., 2025a; Su et al., 2025; Tan et al., 2025; Saunshi et al., 2025; Zhang et al., 2025d) has shown that using latent representations to replace explicit textual reasoning steps allows reasoning models to be more efficient. However, such methods often require extensive-epoch SFT on carefully curated datasets(Hao et al., 2024; Xu et al., 2025d), leading to overfitting on the output format and consequently compromising the model's inherent expressiveness and generalization ability.

**Prompt-based efficient reasoning** methods (Han et al., 2024; Xu et al., 2025b; Lee et al., 2025; Renze & Guven, 2024; Chen et al., 2024) use varying prompts to enforce reasoning models to generate concise CoT with less unnecessary reasoning steps. Especially, (Aytes et al., 2025; Chuang et al., 2024; 2025; Ong et al.) assign different prompts to queries based on their difficulty, thereby adjusting the length of the CoT generated by reasoning models. We also explored the performance of our method combined with prompt design in Tab. 1, demonstrating further reductions in the length of reasoning chains while maintaining considerable accuracy.

Most of the **Output-based efficient reasoning** methods focus on optimizing the best-of-N sampling for LLMs, such as pruning low-quality samples (Xie et al., 2023; Liao et al., 2025) and implementing early stopping (Li et al., 2024; Manvi et al., 2024; Aggarwal et al., 2023) when multiple samples

Table 8: Experimental results on the Qwen3-series models under the officially recommended settings (max_len = 32768, top_p = 0.95, temperature = 0.6).

| Budget | Method | GSM8K | | MATH-500 | | AMC23 | | AIME24 | | AIME25 | | OlympiadB | | Overall | |
|---|---|---|---|---|---|---|---|---|---|---|---|---|---|---|---|
| | | Acc | Tok. | Acc | Tok. | Acc | Tok. | Acc | Tok. | Acc | Tok. | Acc | Tok. | Acc | CR |
| *Qwen3-8B* | | | | | | | | | | | | | | | |
| 32k | *Vanilla* | **95.7** | 2246 | 93.8 | 5368 | 93.8 | 9424 | 70.0 | 16717 | **65.0** | 17880 | **67.9** | 11025 | 81.0 | 100% |
| | *DEER* | 95.5 | **981** | **94.0** | **3227** | **95.0** | **5898** | **75.8** | **12465** | 63.3 | **15135** | 67.0 | **9075** | **81.8** | **68.0%** |
| *Qwen3-14B* | | | | | | | | | | | | | | | |
| 32k | *Vanilla* | 95.7 | 1699 | 94.8 | 4800 | 95.0 | 6837 | **75.0** | 14347 | **76.7** | 16437 | 68.7 | 9992 | 84.3 | 100% |
| | *DEER* | **95.8** | **933** | **95.0** | **3301** | **96.3** | **6299** | 74.2 | **10896** | **76.7** | **15014** | **68.9** | **8263** | **84.5** | **77.6%** |
| *Qwen3-32B* | | | | | | | | | | | | | | | |
| 32k | *Vanilla* | **96.0** | 1714 | **95.8** | 4609 | **98.8** | 7209 | 83.3 | 12874 | **78.3** | 15292 | 69.3 | 9775 | 86.9 | 100% |
| | *DEER* | 95.8 | **992** | 95.4 | **3325** | **98.8** | **5617** | **84.2** | **11531** | **78.3** | **13981** | **69.8** | **8671** | **87.1** | **79.6%** |

Table 9: Accuracy performance on reasoning benchmarks with 95% confidence intervals.

| Model | GSM8K | MATH | AMC23 | AIME24 | GPQA |
|---|---|---|---|---|---|
| Vanilla (ds-7B) | 0.897 [0.891, 0.902] | 0.877 [0.869, 0.884] | 0.794 [0.767, 0.821] | 0.425 [0.400, 0.449] | 0.247 [0.203, 0.291] |
| DEER (ds-7B) | 0.904 [0.896, 0.912] | 0.897 [0.883, 0.911] | 0.856 [0.835, 0.878] | 0.492 [0.463, 0.520] | 0.299 [0.257, 0.341] |
| Vanilla (Qwen3-14B) | 0.948 [0.942, 0.955] | 0.938 [0.932, 0.943] | 0.938 [0.918, 0.957] | 0.708 [0.660, 0.757] | 0.596 [0.571, 0.621] |
| DEER (Qwen3-14B) | 0.955 [0.949, 0.962] | 0.942 [0.936, 0.948] | 0.953 [0.940, 0.967] | 0.754 [0.718, 0.791] | 0.587 [0.566, 0.608] |

achieve self-consistency. However, following the introduction of advanced reasoning models like R1, there is less reliance on best-of-N sampling methods, as these models exhibit strong reasoning capabilities independently. Very recently, two concurrent works share similar motivations with ours. Zhang et al. (2025a) also proposes to terminate early based on trial answers, but requires an additional probe model to determine the correctness. They focus on enhancing the verification capabilities of the probe model, whereas our method explore how to enable the model to self-determine when to exit early and integrate seamlessly into existing reasoning logic. Ma et al. (2025a) prompts reasoning models to directly output final answers during decoding, but only achieves better performance in the low-budget regime or being adapted to best-of-N methods compared to baselines, which limits the applicability and generalization. Song et al. (2025) periodically compresses the KV cache by retaining KV cache that receive high importance score to accelerates inference by leveraging the semantic sparsity of reasoning paths. Jiang et al. (2025b) uses a teacher model to perform skill-aware step decomposition and content pruning, and then distills the pruned reasoning paths into a student model. Huang et al. (2025) projects the steering direction onto the low-dimensional activation manifold and intervenes the activations to reduce thinking tokens.

# N   USE OF LLMS

In the preparation of this manuscript, Large Language Models (LLMs) were employed as auxiliary tools for Language Polishing. During the final stages of manuscript preparation, LLMs were utilized to refine the language of selected passages, including grammar checking, sentence structure optimization, and expression standardization. This process was limited to linguistic improvements and did not involve the generation or modification of any substantive academic content, including research insights, data analysis, or conclusion derivation. It should be emphasized that all core arguments, research methodologies, experimental designs, data analyses, and conclusions presented in this paper were independently developed by the authors. LLMs served solely as language processing aids, and the authors assume full academic responsibility for all content.

Table 10: Token Entropy for *Linguistic Markers* and *Other Tokens*.

| Qwen3-8B | Linguistic Markers | Other Tokens |
|----------|--------------------|--------------|
| gsm8k    | 0.901              | 0.438        |
| math     | 1.058              | 0.385        |
| gpqa     | 1.269              | 0.500        |
| **DS-7B** | **Linguistic Markers** | **Other Tokens** |
| gsm8k    | 1.550              | 0.658        |
| math     | 1.753              | 0.565        |
| gpqa     | 1.241              | 0.510        |

Table 11: Hidden States Cosine Similarity between *Linguistic Markers* and *Other Tokens*.

| Qwen3-8B | Linguistic Markers | Other Tokens |
|----------|--------------------|--------------|
| gsm8k    | 0.262              | 0.543        |
| math     | 0.237              | 0.493        |
| gpqa     | 0.240              | 0.509        |
| **DS-7B** | **Linguistic Markers** | **Other Tokens** |
| gsm8k    | 0.306              | 0.608        |
| math     | 0.247              | 0.530        |
| gpqa     | 0.231              | 0.505        |

Table 12: Confidence interval distribution (%) across tasks for different models.

| Qwen3-14B | 0–0.9 | 0.9–0.97 | 0.97–1.0 |
|-----------|-------|----------|----------|
| gsm8k     | 38.92 | 5.95     | 55.06    |
| math      | 49.53 | 4.83     | 45.46    |
| aime      | 77.20 | 2.45     | 20.35    |
| **Qwen3-8B** | **0–0.9** | **0.9–0.97** | **0.97–1.0** |
| gsm8k     | 35.31 | 5.34     | 59.12    |
| math      | 45.27 | 4.58     | 49.98    |
| aime      | 78.61 | 2.39     | 19.00    |
| **DS-7B** | **0–0.9** | **0.9–0.97** | **0.97–1.0** |
| gsm8k     | 26.54 | 5.29     | 68.17    |
| math      | 34.09 | 5.46     | 60.45    |
| aime      | 80.90 | 1.41     | 17.69    |

Table 13: Experimental results on more types of reasoning models and reasoning benchmarks. "Acc" denotes accuracy, "Tok" denotes token count, and "CR" denotes compression rate. ↑ indicates that higher values are better, while ↓ indicates that lower values are better. The best results are highlighted in **bold**.

| Method | GSM8K | | | MATH-500 | | | MATH / AMC23 | | | AIME24 | | | AIME25 | | | OlympiadBench | | | SCIENCE / GPQA-D | | | Overall | |
|---|---|---|---|---|---|---|---|---|---|---|---|---|---|---|---|---|---|---|---|---|---|---|---|
| | Acc↑ | Tok↓ | CR↓ | Acc↑ | Tok↓ | CR↓ | Acc↑ | Tok↓ | CR↓ | Acc↑ | Tok↓ | CR↓ | Acc↑ | Tok↓ | CR↓ | Acc↑ | Tok↓ | CR↓ | Acc↑ | Tok↓ | CR↓ | Acc↑ | CR↓ |
| ***DeepSeek-R1-Distill-Qwen-32B*** | | | | | | | | | | | | | | | | | | | | | | | |
| Vanilla | 94.3 | 1,202 | 100% | 89.2 | 3,736 | 100% | 87.5 | 5,354 | 100% | 56.7 | 10,293 | 100% | 43.3 | 11,075 | 100% | 55.7 | 7,334 | 100% | 56.1 | 7,181 | 100% | 69.0 | 100% |
| DEER | 95.1 | 819 | 68.1% | 90.4 | 2,425 | 64.9% | 95 | 4,252 | 79.4% | 63.3 | 7,424 | 72.1% | 46.7 | 8,913 | 80.5% | 57.8 | 5,351 | 73.0% | 64.1 | 4,943 | 68.8% | **73.2** | **72.4%** |
| ***DeepSeek-R1-Distill-Qwen-14B*** | | | | | | | | | | | | | | | | | | | | | | | |
| Vanilla | 93.9 | 1,458 | 100% | 88.6 | 3,815 | 100% | 82.5 | 6,545 | 100% | 51.7 | 11,211 | 100% | 36.7 | 12,304 | 100% | 52.6 | 7,908 | 100% | 52.0 | 6,731 | 100% | 65.4 | 100% |
| DEER | 93.3 | 1,040 | 71.3% | 89.8 | 2,577 | 67.5% | 85.0 | 4,240 | 64.8% | 68.4 | 8,115 | 72.4% | 36.7 | 10,125 | 82.3% | 55.0 | 5,736 | 72.5% | 56.6 | 4,856 | 72.1% | **69.3** | **71.9%** |
| ***DeepSeek-R1-Distill-Qwen-7B*** | | | | | | | | | | | | | | | | | | | | | | | |
| Vanilla | 89.6 | 1,484 | 100% | 87.4 | 3,858 | 100% | 78.8 | 6,792 | 100% | 41.7 | 13,765 | 100% | 26.7% | 12,767 | 100% | 47.3 | 8,563 | 100% | 23.7 | 10,247 | 100% | 56.5 | 100% |
| DEER | 90.6 | 917 | 61.8% | 89.8 | 2,143 | 55.5% | 85.0 | 4,451 | 65.5% | 49.2 | 9,839 | 71.5% | 36.7 | 7,257 | 56.8% | 52.6 | 5,420 | 63.3% | 31.3 | 5,469 | 53.4% | **62.2** | **61.1%** |
| ***DeepSeek-R1-Distill-Qwen-1.5B*** | | | | | | | | | | | | | | | | | | | | | | | |
| Vanilla | 76.1 | 1,617 | 100% | 69.0 | 6,018 | 100% | 52.5 | 8,819 | 100% | 23.3 | 13,702 | 100% | 13.3 | 14,450 | 100% | 28.0 | 11,200 | 100% | 7.1 | 13,029 | 100% | 38.5 | 100% |
| DEER | 74.7 | 984 | 60.9% | 67.8 | 2,497 | 41.5% | 60.0 | 5,496 | 62.3% | 23.3 | 9,557 | 69.7% | 10.0 | 9,281 | 64.2% | 32.0 | 5,960 | 53.2% | 12.1 | 5,762 | 44.2% | **40.0** | **56.6%** |
| ***Qwen3-32B*** | | | | | | | | | | | | | | | | | | | | | | | |
| Vanilla | 96.3 | 1,668 | 100% | 94.4 | 4,440 | 100% | 95.0 | 7,627 | 100% | 73.3 | 11,374 | 100% | 65.0 | 12,446 | 100% | 63.4 | 6,438 | 100% | 65.2 | 6,893 | 100% | 78.9 | 100% |
| DEER | 96.2 | 769 | 46.1% | 94.2 | 3,418 | 77.0% | 97.5 | 5,753 | 75.4% | 76.7 | 8,682 | 76.3% | 66.7 | 10,893 | 87.5% | 67.9 | 5,189 | 80.6% | 64.7 | 4,167 | 60.5% | **80.6** | **71.9%** |
| ***Qwen3-14B*** | | | | | | | | | | | | | | | | | | | | | | | |
| Vanilla | 95.1 | 2,047 | 100% | 93.8 | 4,508 | 100% | 95.0 | 7,139 | 100% | 70.0 | 10,859 | 100% | 63.3 | 12,286 | 100% | 62.5 | 8,692 | 100% | 60.1 | 7,339 | 100% | 77.1 | 100% |
| DEER | 95.3 | 840 | 41.0% | 94.0 | 3,074 | 68.2% | 95.0 | 4,763 | 66.7% | 76.7 | 7,619 | 70.2% | 66.7 | 11,135 | 90.6% | 67.4 | 7,060 | 81.2% | 57.6 | 2,898 | 39.5% | **79.0** | **65.0%** |
| ***Qwen3-8B*** | | | | | | | | | | | | | | | | | | | | | | | |
| Vanilla | 94.9 | 2,245 | 100% | 91.2 | 5,216 | 100% | 87.5 | 7,986 | 100% | 65.0 | 12,110 | 100% | 54.2 | 12,835 | 100% | 59.3 | 9,487 | 100% | 51.5 | 9,145 | 100% | 71.9 | 100% |
| DEER | 95.2 | 1,071 | 47.7% | 92.6 | 2,732 | 52.4% | 92.5 | 4,392 | 55.0% | 61.7 | 8,796 | 72.6% | 60.0 | 12,229 | 95.3% | 62.4 | 7,479 | 78.8% | 52.5 | 3,111 | 34.0% | **73.8** | **62.3%** |
| ***Qwen3-4B*** | | | | | | | | | | | | | | | | | | | | | | | |
| Vanilla | 94.1 | 2,175 | 100% | 92.2 | 4,767 | 100% | 87.5 | 7,443 | 100% | 63.3 | 11,916 | 100% | 48.4 | 13,112 | 100% | 59.3 | 9,098 | 100% | 46.5 | 9,294 | 100% | 70.2 | 100% |
| DEER | 94.5 | 1,250 | 57.5% | 92.6 | 3,214 | 67.4% | 87.5 | 4,906 | 65.9% | 63.3 | 9,327 | 78.3% | 55.0 | 12,039 | 91.8% | 64.7 | 7,569 | 83.2% | 47.5 | 3,275 | 35.4% | **72.2** | **68.5%** |
| ***Qwen3-1.7B*** | | | | | | | | | | | | | | | | | | | | | | | |
| Vanilla | 90.1 | 2,045 | 100% | 85.6 | 5,160 | 100% | 70.0 | 8,637 | 100% | 30.0 | 13,758 | 100% | 26.7 | 13,943 | 100% | 52.2 | 9,437 | 100% | 35.9 | 9,271 | 100% | 55.8 | 100% |
| DEER | 90.3 | 1,066 | 52.1% | 85.6 | 2,463 | 47.7% | 70.0 | 4,673 | 54.1% | 30.0 | 7,943 | 57.7% | 36.7 | 11,579 | 83.0% | 52.6 | 7,257 | 76.9% | 43.4 | 3,549 | 38.3% | **58.4** | **58.6%** |
| ***QwQ-32B*** | | | | | | | | | | | | | | | | | | | | | | | |
| Vanilla | 96.7 | 1,427 | 100% | 93.8 | 4,508 | 100% | 92.5 | 6,792 | 100% | 66.7 | 10,821 | 100% | 46.7 | 12,300 | 100% | 65.2 | 8,546 | 100% | 63.1 | 7,320 | 100% | 75.0 | 100% |
| DEER | 96.3 | 977 | 68.5% | 94.6 | 3,316 | 73.6% | 95.0 | 5,782 | 85.1% | 70.0 | 10,097 | 93.3% | 50.0 | 11,598 | 94.3% | 65.2 | 7,639 | 89.4% | 64.1 | 6,163 | 84.2% | **76.5** | **84.0%** |
| ***Llama-3.1-Nemotron-Nano-8B-v1*** | | | | | | | | | | | | | | | | | | | | | | | |
| Vanilla | 89.2 | 1,618 | 100% | 91.2 | 3,794 | 100% | 90.0 | 6,153 | 100% | 56.7 | 10,821 | 100% | 35.0 | 13,192 | 100% | 54.3 | 7,321 | 100% | 41.9 | 8,074 | 100% | 65.5 | 100% |
| DEER | 89.8 | 1,473 | 91.0% | 91.4 | 2,995 | 78.9% | 90.0 | 5,408 | 87.9% | 66.7 | 9,755 | 90.6% | 36.7 | 11,820 | 89.6% | 60.7 | 6,407 | 87.5% | 47.5 | 7,576 | 93.8% | **69.0** | **88.5%** |

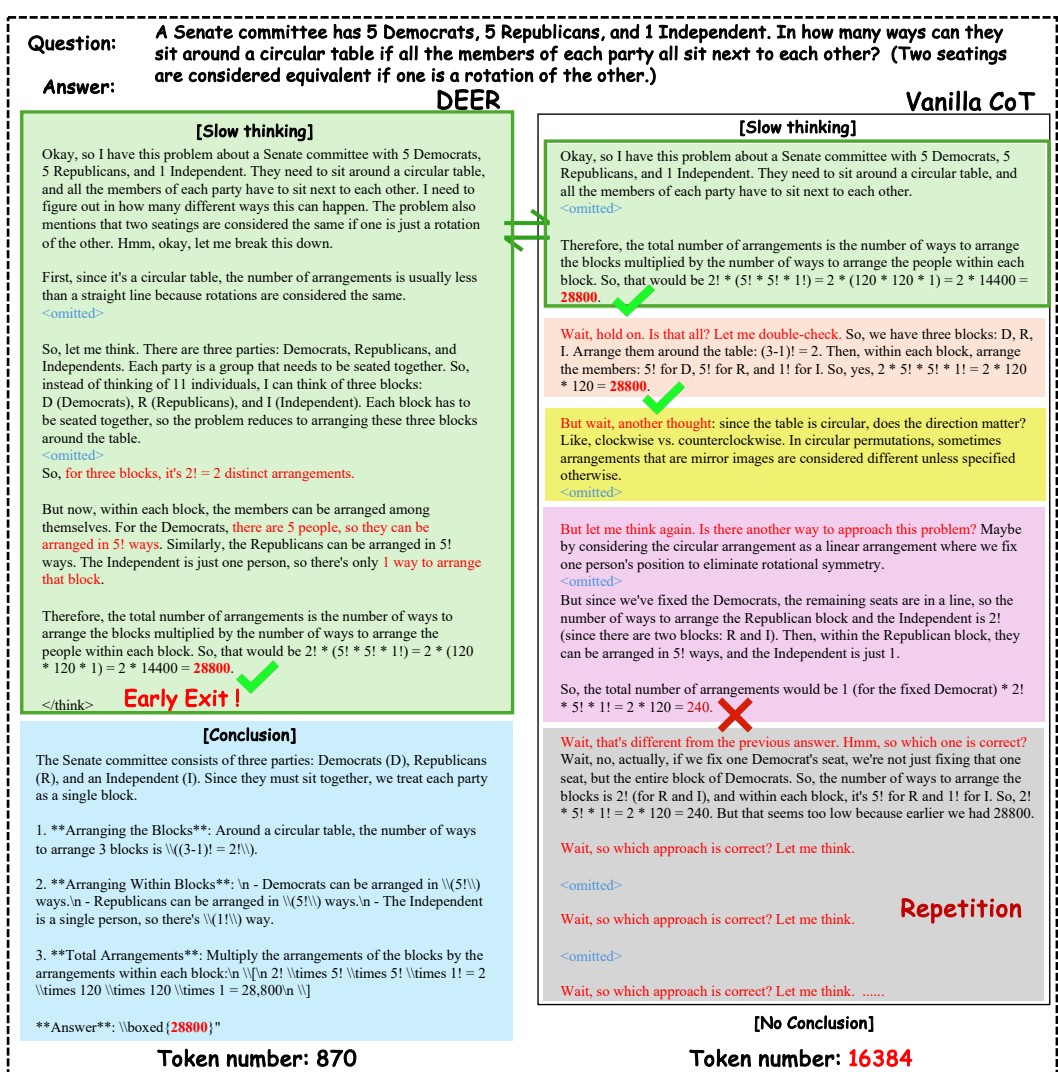

Figure 13: Comparison of generated content samples between DEER and Vanilla CoT on MATH-500. Both DEER and vanilla CoT arrive at the correct answer during the first reasoning step, as shown in the green box. The difference lies in the fact that DEER exits early after evaluating the confidence of the trial answer as sufficiently high, thus producing the correct result.

**Question:** The coordinates of a parallelogram are (5, 3), (6, 8), (7, 4) and $(x, y)$ and $x > 7$. What is the value of $x + y$?

**Answer:**

**DEER**

Okay, so I have this problem where I need to find the coordinates of the fourth vertex of a parallelogram. The given points are (5, 3), (6, 8), (7, 4), and (x, y) with the condition that x > 7. Then, I need to find the value of x + y. Hmm, let me think about how to approach this.\n\nFirst, I remember that in a parallelogram, opposite sides are equal and parallel. That means the midpoints of the diagonals should be the same. So, if I can find the midpoint of one diagonal, it should be equal to the midpoint of the other diagonal.\n\nLet me label the points to make it clearer. Let's say point A is (5, 3), point B is (6, 8), point C is (7, 4), and point D is (x, y). Now, I need to figure out which points are connected to each other. Since it's a parallelogram, the order of the points matters. I think the points should be connected in order, so A to B to C to D and back to A. But I'm not entirely sure, so maybe I should consider different possibilities.

<omitted>

Let me try another approach.\n\nAnother property of a parallelogram is that the diagonals bisect each other. So, the midpoint of diagonal AC should be the same as the midpoint of diagonal BD.\n\nLet me calculate the midpoint of AC first. Point A is (5, 3) and point C is (7, 4). The midpoint formula is ((x1 + x2)/2, (y1 + y2)/2). So, midpoint of AC is ((5 + 7)/2, (3 + 4)/2) = (12/2, 7/2) = (6, 3.5).

<omitted>

Midpoint is ((5 + x)/2, (3 + y)/2).\n\nMidpoint of BC: Point B is (6, 8) and point C is (7, 4). Midpoint is ((6 + 7)/2, (8 + 4)/2) = (13/2, 12/2) = (6.5, 6).\n\nSetting midpoints equal:\n\n(5 + x)/2 = 6.5 and (3 + y)/2 = 6.\n\nSolving the first equation: (5 + x)/2 = 6.5 => 5 + x = 13 => x = 8.\n\nSolving the second equation: (3 + y)/2 = 6 => 3 + y = 12 => y = 9.\n\nOkay, this gives x = 8 and y = 9, which satisfies x > 7. So, the fourth point is (8, 9), and x + y = 8 + 9 = 17.

</think>

To find the coordinates of the fourth vertex of the parallelogram, we use the property that the diagonals of a parallelogram bisect each other. \n\n1. **Calculate the midpoint of diagonal BC:**\n   - Points B (6, 8) and C (7, 4).\n   - Midpoint formula: $\left(\frac{6 + 7}{2}, \frac{8 + 4}{2}\right) = \left(\frac{13}{2}, 6\right) = (6.5, 6)$.\n\n2. **Set the midpoint of diagonal AD equal to the midpoint of BC:**\n   - Let D be (x, y).\n   - Midpoint of AD: $\left(\frac{5 + x}{2}, \frac{3 + y}{2}\right)$.\n   - Equate to (6.5, 6):\n     - $\frac{5 + x}{2} = 6.5 \Rightarrow x = 8$.\n     - $\frac{3 + y}{2} = 6 \Rightarrow 3 + y = 12 \Rightarrow y = 9$.\n\n3. **Calculate $x + y$:**\n   - $x + y = 8 + 9 = 17$.\n\n**Answer:** $\boxed{17}$

**Token number: ~800**

**Vanilla CoT**

Okay, so I have this problem where I need to find the coordinates of the fourth vertex of a parallelogram. The given points are (5, 3), (6, 8), (7, 4), and (x, y) with the condition that x > 7. Then, I need to find the value of x + y. Hmm, let me think about how to approach this.\n\nFirst, I remember that in a parallelogram, opposite sides are equal and parallel. That means the midpoints of the diagonals should be the same. So, if I can find the midpoint of one diagonal, it should be equal to the midpoint of the other diagonal.\n\nLet me label the points to make it clearer. Let's say point A is (5, 3), point B is (6, 8), point C is (7, 4), and point D is (x, y). Now, I need to figure out which points are connected to each other. Since it's a parallelogram, the order of the points matters. I think the points should be connected in order, so A to B to C to D and back to A. But I'm not entirely sure, so maybe I should consider different possibilities.

Let me try another approach.\n\nAnother property of a parallelogram is that the diagonals bisect each other. So, the midpoint of diagonal AC should be the same as the midpoint of diagonal BD.

<omitted>

Midpoint is ((5 + x)/2, (3 + y)/2).\n\nMidpoint of BC: Point B is (6, 8) and point C is (7, 4). Midpoint is ((6 + 7)/2, (8 + 4)/2) = (13/2, 12/2) = (6.5, 6).\n\nSetting midpoints equal:\n\n(5 + x)/2 = 6.5 and (3 + y)/2 = 6.\n\nSolving the first equation: (5 + x)/2 = 6.5 => 5 + x = 13 => x = 8.\n\nSolving the second equation: (3 + y)/2 = 6 => 3 + y = 12 => y = 9.\n\nOkay, this gives x = 8 and y = 9, which satisfies x > 7. So, the fourth point is (8, 9), and x + y = 8 + 9 = 17.

**Wait, let me verify this.** If the points are A(5,3), B(6,8), C(7,4), and D(8,9), does this form a parallelogram?\n\nLet me check the vectors.

<omitted>

These aren't equal either. Hmm, that's confusing.\n\nWait, maybe I made a mistake in assigning the points. Let me try another approach.

<omitted>

Hmm, these aren't equal. Wait, that's a problem.\n\nWait, but according to the vector addition, D should be (8,9). So, why are vectors AD and BC not equal?\n\nWait, maybe I need to check the other pair of sides. Vector AD and vector BC are not equal, but vector AB and vector DC are equal.

<omitted>

These aren't equal, but in a parallelogram, both pairs should be equal. So, that's a problem.\n\nWait, but according to the midpoint method, it's a parallelogram. So, maybe I'm missing something.

<omitted>

**No answer !**

**Token number: 16384**

Figure 14: Comparison of generated content between DEER and Vanilla on MATH-500.

**Question:** Point $A$ lies somewhere within or on the square which has opposite corners at $(0,0)$ and $(2,2)$. Point $B$ lies somewhere within or on the square which has opposite

**Answer:** corners at points $(4,2)$ and $(5,3)$. What is the greatest possible value of the slope of the line containing points $A$ and $B$? Express your answer as a common fraction.

### DEER

Okay, so I have this problem where I need to find the greatest possible slope of a line connecting two points, A and B. Point A is somewhere inside or on a square with opposite corners at (0,0) and (2,2). Point B is inside or on another square with opposite corners at (4,2) and (5,3). I need to figure out the maximum slope possible between any two points A and B in these squares.\n\nFirst, let me visualize the squares. The first square has corners at (0,0) and (2,2). Similarly, the second square has opposite corners at (4,2) and (5,3). Hmm, so that square is a bit smaller. Let me figure out its sides. The distance between (4,2) and (5,3) is sqrt[(5-4)^2 + (3-2)^2] = sqrt[1 + 1] = sqrt(2). So, it's a square with side length sqrt(2), rotated 45 degrees relative to the axes.\n\nWait, actually, no. If the opposite corners are (4,2) and (5,3), then the sides are not aligned with the axes. <omitted>
But wait, is that achievable? Because point A is in the first square, so if A is at (2,0), which is a corner, and point B is at (4,3), which is a corner of the second square. So, connecting (2,0) to (4,3) would give a slope of (3 - 0)/(4 - 2) = 3/2. But is that the maximum?

\</think\>
To find the greatest possible slope of the line containing points \\( A \\) and \\( B \\), we need to maximize the expression \\( \\frac{y_B - y_A}{x_B - x_A} \\). \n\n- Point \\( A \\) lies within or on the square with opposite corners at \\( (0,0) \\) and \\( (2,2) \\). The minimum \\( y \\)-coordinate for \\( A \\) is \\( 0 \\) and the maximum \\( x \\)-coordinate is \\( 2 \\).\n- Point \\( B \\) lies within or on the square with opposite corners at \\( (4,2) \\) and \\( (5,3) \\). The maximum \\( y \\)-coordinate for \\( B \\) is \\( 3 \\) and the minimum \\( x \\)-coordinate is \\( 4 \\).\n\nTo maximize the slope, we should choose \\( A \\) at \\( (2,0) \\) and \\( B \\) at \\( (4,3) \\). The slope is then:\n\\[\n\\text{slope} = \\frac{3 - 0}{4 - 2} = \\frac{3}{2}\n\\]\n\nHowever, upon closer inspection, the square for \\( B \\) is actually a diamond with corners at \\( (4,2) \\), \\( (5,3) \\), \\( (5,2) \\), and \\( (4,3) \\). The point \\( (4,3) \\) is indeed a corner of this square. Therefore, the maximum slope is achievable.\n\nThe greatest possible slope is \\( \\frac{3}{2} \\).\n\n\\boxed{\\dfrac{3}{2}}"], "gold_answer": "\\frac{3}{2}"}

**Token number: ~1000**

### Vanilla CoT

Okay, so I have this problem where I need to find the greatest possible slope of a line connecting two points, A and B. Point A is somewhere inside or on a square with opposite corners at (0,0) and (2,2). Point B is inside or on another square with opposite corners at (4,2) and (5,3). I need to figure out the maximum slope possible between any two points A and B in these squares.\n\nFirst, let me visualize the squares. The first square has corners at (0,0) and (2,2). Similarly, the second square has opposite corners at (4,2) and (5,3). Hmm, so that square is a bit smaller. Let me figure out its sides. The distance between (4,2) and (5,3) is sqrt[(5-4)^2 + (3-2)^2] = sqrt[1 + 1] = sqrt(2). So, it's a square with side length sqrt(2), rotated 45 degrees relative to the axes.\n\nWait, actually, no. If the opposite corners are (4,2) and (5,3), then the sides are not aligned with the axes. <omitted>
But wait, is that achievable? Because point A is in the first square, so if A is at (2,0), which is a corner, and point B is at (4,3), which is a corner of the second square. So, connecting (2,0) to (4,3) would give a slope of (3 - 0)/(4 - 2) = 3/2. But is that the maximum?

**Wait, but maybe we can get a steeper slope by choosing points that are not at the corners.** Because if we can have a larger yB - yA with a smaller xB - xA, that would give a steeper slope.
<omitted>
**Wait, but hold on. Let me think again.** Maybe if we don't take A at (2,0) and B at (4,3), but somewhere else, we can get a steeper slope.
<omitted>
But wait, is there a way to have a larger slope? For example, if A is at (2,2) and B is at (4,3). Then, the slope is (3 - 2)/(4 - 2) = 1/2, which is less than 3/2.\n\nAlternatively, if A is at (0,0) and B is at (5,3), then the slope is (3 - 0)/(5 - 0) = 3/5, which is less than 3/2.
<omitted>
Alternatively, if point A is at (x, yA) where x < 2, then point B would have to be at (x + 2, yB). But since the second square starts at x=4, x + 2 must be at least 4, so x must be at least 2.
<omitted>

**No answer !**

**Token number: 16384**

Figure 15: Comparison of generated content between DEER and Vanilla on MATH-500.

**Question:** The expression $$1 + 2 + 3 - 4 + 5 + 6$$ is equivalent to 13. If we add parentheses in various locations, we can change this value. For example, $1+2+3-(4+5)+6=1+2+3-9+6=6-9+6=3$. If we only add parentheses (and not rearrange the order of the terms), what is the minimum possible value this expression can obtain? (For the purposes of this problem, we cannot change addition or subtraction into multiplication. For example, $(1+2+3)(-4)+5+6$ is not a valid placement of parentheses.)\n\nEnter your answer as a single value.",

**Answer:**

**DEER**

Okay, so I have this math problem here: the expression is 1 + 2 + 3 - 4 + 5 + 6, and it's equal to 13. The task is to add parentheses in various places to change the value, but I can't rearrange the terms or turn the operations into multiplication. I need to find the minimum possible value this expression can obtain by adding parentheses. Hmm, interesting.\n\nFirst, let me make sure I understand the problem correctly. I can only add parentheses, which will change the order in which operations are performed, but I can't move the numbers around or change the plus and minus signs into multiplication or division. So, for example, I can do something like (1 + 2) + 3 - 4 + 5 + 6, but I can't do 1*(2+3) - 4 + 5 + 6 because that would involve multiplication, which isn't allowed.
<omitted>
Yep, that's correct. So, 13 is the original value.\n\nAlternatively, group (1 + 2) + (3 - 4 + 5 + 6). So, 1 + 2 = 3, 3 - 4 = -1, -1 + 5 = 4, 4 + 6 = 10. Then, 3 + 10 = 13.\n\nAlternatively, group 1 + (2 + 3 - 4 + 5) + 6. So, 2 + 3 = 5, 5 - 4 = 1, 1 + 5 = 6. Then, 1 + 6 + 6 = 13.\n\nHmm, seems like 3 is the lowest I can get so far. But let me think if there's another way.\n\nWhat if I group 1 + 2 + 3 - (4 + 5 + 6). Let's compute that: 4 + 5 + 6 = 15. Then, 1 + 2 + 3 = 6, 6 - 15 = -9. So, that's -9. Wait, that's way lower than 3. Is that allowed?

`</think>`

To find the minimum possible value of the expression $\(1 + 2 + 3 - 4 + 5 + 6\)$ by adding parentheses, we can strategically group terms to maximize the impact of the subtraction. \n\nBy grouping the terms after the subtraction, we can create a larger number to subtract, significantly reducing the overall value. For example, grouping $\(4 + 5 + 6\)$ together:\n\n\[\n1 + 2 + 3 - (4 + 5 + 6) = 6 - 15 = -9\n\]\n\nThus, the minimum possible value is **\(\\boxed{-9}\)**

**Token number: ~1500**

**Vanilla CoT**

Okay, so I have this math problem here: the expression is 1 + 2 + 3 - 4 + 5 + 6, and it's equal to 13. The task is to add parentheses in various places to change the value, but I can't rearrange the terms or turn the operations into multiplication. I need to find the minimum possible value this expression can obtain by adding parentheses. Hmm, interesting.\n\nFirst, let me make sure I understand the problem correctly. I can only add parentheses, which will change the order in which operations are performed, but I can't move the numbers around or change the plus and minus signs into multiplication or division. So, for example, I can do something like (1 + 2) + 3 - 4 + 5 + 6, but I can't do 1*(2+3) - 4 + 5 + 6 because that would involve multiplication, which isn't allowed.
<omitted>
Yep, that's correct. So, 13 is the original value.\n\nAlternatively, group (1 + 2) + (3 - 4 + 5 + 6). So, 1 + 2 = 3, 3 - 4 = -1, -1 + 5 = 4, 4 + 6 = 10. Then, 3 + 10 = 13.\n\nAlternatively, group 1 + (2 + 3 - 4 + 5) + 6. So, 2 + 3 = 5, 5 - 4 = 1, 1 + 5 = 6. Then, 1 + 6 + 6 = 13.\n\nHmm, seems like 3 is the lowest I can get so far. But let me think if there's another way.\n\nWhat if I group 1 + 2 + 3 - (4 + 5 + 6). Let's compute that: 4 + 5 + 6 = 15. Then, 1 + 2 + 3 = 6, 6 - 15 = -9. So, that's -9. Wait, that's way lower than 3. Is that allowed?

**Wait, hold on.** The original expression is 1 + 2 + 3 - 4 + 5 + 6. If I group 4 + 5 + 6, that would be subtracting 15, so 6 - 15 = -9.
<omitted>
**Wait,** but in the original expression, after the subtraction, it's +5 +6. So, if I group 4 +5 +6, that would be changing the operations after the subtraction, but I can't rearrange terms. So, is that allowed?
<omitted>
Alternatively, group 1 + (2 +3 -4 +5 +6). As before, 1 +12=13.\n\nAlternatively, group (1 +2 +3 -4 +5) +6. ...

**Endless exhaustive search!**

**Token number: 16384**

Figure 16: Comparison of generated content between DEER and Vanilla on MATH-500.

**Question:** The coordinates of a parallelogram are (5, 3), (6, 8), (7, 4) and $(x, y)$ and $x > 7$. What is the value of $x + y$?

**Answer:**

Okay, so I have this problem where I need to find the coordinates of the fourth vertex of a parallelogram. The given points are (5, 3), (6, 8), (7, 4), and (x, y) with the condition that x > 7. Then, I need to find the value of x + y. Hmm, let me think about how to approach this.

First, I remember that in a parallelogram, opposite sides are equal and parallel. That means the midpoints of the diagonals should be the same. So, if I can find the midpoint of one diagonal, it should be equal to the midpoint of the other diagonal.

Let me label the points to make it clearer. Let's say point A is (5, 3), point B is (6, 8), point C is (7, 4), and point D is (x, y). Now, I need to figure out which points are connected to each other. Since it's a parallelogram, the order of the points matters. I think the points should be connected in order, so A to B to C to D and back to A. But I'm not entirely sure, so maybe I should consider different possibilities.

Alternatively, another way to approach this is by using vectors. In a parallelogram, the vector from one point to another should be equal to the vector from another point to the next. So, for example, vector AB should be equal to vector DC, and vector AD should be equal to vector BC.

Let me try that. First, let's find vector AB. Point A is (5, 3) and point B is (6, 8). So, vector AB is (6 - 5, 8 - 3) = (1, 5). If vector AB is equal to vector DC, then vector DC should also be (1, 5). Point D is (x, y) and point C is (7, 4). So, vector DC is (7 - x, 4 - y). Setting this equal to (1, 5), we get:\n\n7 - x = 1 and 4 - y = 5.

Solving the first equation: 7 - x = 1 => x = 7 - 1 = 6.

Solving the second equation: 4 - y = 5 => y = 4 - 5 = -1.

**<Induced trial answer>:** \boxed{6 + (-1)}. ❌   Confidence: 0.7344     Continue thinking

**Wait,** but the problem states that x > 7, and here x is 6, which is less than 7. That can't be right. So, maybe I assigned the points incorrectly.

Perhaps the order of the points isn't A, B, C, D. Maybe it's a different order. Let me try another approach.

Another property of a parallelogram is that the diagonals bisect each other. So, the midpoint of diagonal AC should be the same as the midpoint of diagonal BD.

Let me calculate the midpoint of AC first. Point A is (5, 3) and point C is (7, 4). The midpoint formula is ((x1 + x2)/2, (y1 + y2)/2). So, midpoint of AC is ((5 + 7)/2, (3 + 4)/2) = (12/2, 7/2) = (6, 3.5).

Now, the midpoint of BD should also be (6, 3.5). Point B is (6, 8) and point D is (x, y). So, midpoint of BD is ((6 + x)/2, (8 + y)/2). Setting this equal to (6, 3.5):\n\n(6 + x)/2 = 6 and (8 + y)/2 = 3.5.

Solving the first equation: (6 + x)/2 = 6 => 6 + x = 12 => x = 6.\n\nSolving the second equation: (8 + y)/2 = 3.5 => 8 + y = 7 => y = -1.

Again, I get x = 6 and y = -1, but x is supposed to be greater than 7. Hmm, that's conflicting. Maybe I need to consider a different pair of points as the diagonal.

**<Induced trial answer>:** \boxed{10}. ❌   Confidence: 0.5312     Continue thinking

**Wait,** perhaps I assumed the wrong diagonal. Maybe the diagonals are AD and BC instead of AC and BD. Let me try that.

Midpoint of AD: Point A is (5, 3) and point D is (x, y). Midpoint is ((5 + x)/2, (3 + y)/2).

Midpoint of BC: Point B is (6, 8) and point C is (7, 4). Midpoint is ((6 + 7)/2, (8 + 4)/2) = (13/2, 12/2) = (6.5, 6).

Setting midpoints equal:\n\n(5 + x)/2 = 6.5 and (3 + y)/2 = 6.

Solving the first equation: (5 + x)/2 = 6.5 => 5 + x = 13 => x = 8.\n\nSolving the second equation: (3 + y)/2 = 6 => 3 + y = 12 => y = 9.

Okay, this gives x = 8 and y = 9, which satisfies x > 7. So, the fourth point is (8, 9), and x + y = 8 + 9 = 17.

**<Induced trial answer>:** \boxed{17}. ✅   Confidence: 0. 9993    Exit!

**</think>**

To find the coordinates of the fourth vertex of the parallelogram, we use the property that the diagonals of a parallelogram bisect each other.

1.  **Calculate the midpoint of diagonal BC:**\n - Points B (6, 8) and C (7, 4).\n - Midpoint formula: \\(\\left(\\frac{6 + 7}{2}, \\frac{8 + 4}{2}\\right) = \\left(\\frac{13}{2}, 6\\right) = (6.5, 6)\\).

2.  2. **Set the midpoint of diagonal AD equal to the midpoint of BC:**\n - Let D be (x, y).\n - Midpoint of AD: \\(\\left(\\frac{5 + x}{2}, \\frac{3 + y}{2}\\right)\\).\n - Equate to (6.5, 6):\n - \\(\\frac{5 + x}{2} = 6.5 \\Rightarrow 5 + x = 13 \\Rightarrow x = 8\\).\n - \\(\\frac{3 + y}{2} = 6 \\Rightarrow 3 + y = 12 \\Rightarrow y = 9\\).

3.  3. **Calculate \\(x + y\\):**\n - \\(x + y = 8 + 9 = 17\\).

4.  **Answer:** \\(\\boxed{17}\\)

Figure 17: An example demonstrating LRM's implicit ability to sense the emergence of pearl reasoning. When the model's reasoning content is insufficient to support the elicited answer, the confidence remains at a low level, as demonstrated by the first two confidence values in the figure. Conversely, when the model has provided adequate and sound reasoning as justification, the confidence reaches high levels, potentially approaching 1, as shown by the final confidence value in the figure.

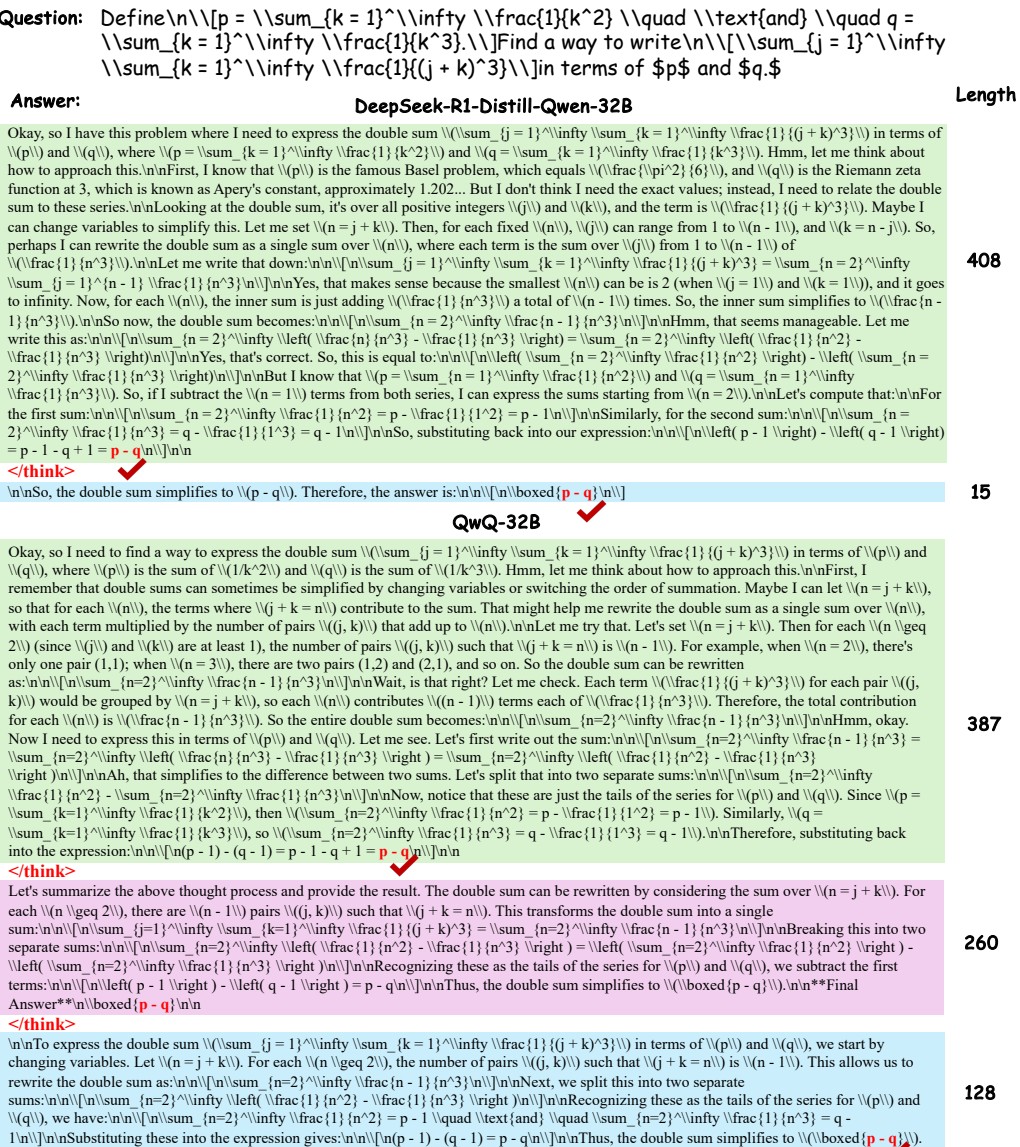

Figure 18: An example demonstrating the presence of more than one ⟨/think⟩ tokens in QwQ-32B. For both models, the green thinking chunk is sufficient to draw the correct conclusion. However, QwQ-32B proceeds with an additional summary (red chunk) and generates its own ⟨/think⟩ token. Based on all the above content, it arrives at the conclusion.

