# OpenReview forum: "Dynamic Early Exit in Reasoning Models"
_ICLR.cc/2026/Conference — ICLR 2026 Poster_

### Official Review · Reviewer_Ax2Q · 2025-10-27

**Soundness:** 3
**Presentation:** 3
**Contribution:** 3
**Rating:** 6
**Confidence:** 3

**Summary:**

Summary

This paper proposes DEER (Dynamic Early Exit in Reasoning), a training-free framework that allows large reasoning models (LRMs) to dynamically terminate the chain-of-thought (CoT) generation when the model is already confident enough in its answer. The method monitors reasoning transition points through linguistic or entropy-based signals, induces trial answers at these points, and uses token-level confidence scores to decide whether to stop reasoning early. The authors also introduce DEER-Pro, a robust variant using multiple answer inducers and calibrated confidence aggregation.
Experiments across 10 reasoning benchmarks (math, science, and programming) and 11 models (1.5B–671B parameters) show substantial token reduction (19–80%) and accuracy improvements (up to +5%). The method is efficient, plug-and-play, and demonstrates strong robustness to thresholds and model scales.

**Strengths:**

Clear Motivation & Insightful Observation:
The paper convincingly identifies the “overthinking” problem in modern reasoning LLMs and empirically verifies the existence of pearl reasoning — points where further CoT expansion is redundant or even harmful.

Simple Yet Effective Approach:
DEER is training-free, easy to integrate, and works well across different reasoning models. It leverages confidence signals during inference — a practical and theoretically grounded design.

Comprehensive Experiments & Strong Results:
Evaluations cover diverse tasks (GSM8K, AIME, GPQA, BigCodeBench) and model families. DEER consistently outperforms baselines such as CoD, Dynasor-CoT, and SEAL in both accuracy and efficiency.

**Weaknesses:**

The paper uses a limited number of backbone models, and it remains unclear how well the proposed method performs on closed-source reasoning models.

Mathematical reasoning benchmarks typically use pass@k metrics to reduce randomness and improve robustness, but this paper does not adopt that evaluation standard.

The authors do not include ablation studies to verify how each component or module of the proposed method contributes to the overall performance improvement.

**Questions:**

None

---

> ### Author Response · Authors · 2025-11-19
> **Response to Reviewer Ax2Q (1/2)**
>
> We sincerely thank you for your thorough and insightful evaluation of our work. We are particularly grateful for the positive assessment of our paper's core strengths.
> We are very encouraged that the reviewer found our motivation to be **"clear and insightful,"** our DEER approach **"simple yet effective,"** and our experimental results **"comprehensive and strong."**  Below, we address each of your concerns in detail.
>
> ## Response to weakness 1:  Performance on closed-source reasoning models
>
> We thank the reviewer for the feedback on backbone model diversity.
>
> To clarify, our experiments covered **11 different backbone models ranging from 1.5B to 671B parameters, encompassing most commonly used LRMs to the best of our ability.**
>
> Regarding closed-source models, we agree that testing DEER on these models would strengthen our generalization claims. Regarding this concern, we provide responses from the following two perspectives:
>
> - **Feasibility and Value for Closed-Source Models**: The current limitation arises from API access restrictions rather than technical incompatibility. While these **APIs currently withhold prediction probabilities for security reasons**, DEER's core mechanism of monitoring token-level confidence remains a standard feature inherent in autoregressive models like GPT and Gemini.
> We believe DEER offers clear benefits to model providers. As a plug-and-play solution, it can reduce computational costs, lower latency, and increase throughput with minimal integration effort. These improvements align directly with the key challenges in large-scale model deployment.
>
> - **Evidence from Large-Scale Open-Source Models**: While we cannot directly test on closed-source models, we evaluated DEER on the largest available open-source alternatives. Our results on the **671B DeepSeek-R1 model (Appendix K)** demonstrate that DEER scales effectively to models of comparable size, providing strong evidence for its generalizability.
>
>
> ## Response to weakness 2: Why use pass@1 not pass@k
>
> We appreciate the reviewer's suggestion regarding the pass@k metric.
>
> To clarify our choice of metric, we would like to briefly discuss the distinction between pass@1 and pass@k in the context of our research goals. The **pass@k** metric evaluates the probability of finding at least one correct solution among k sampled outputs, which is typically used to **measure a model's upper-bound capabilities by increasing computational budget**. In contrast, our research focuses on optimizing the efficiency and accuracy of a single reasoning pass. Therefore, we chose **pass@1** (i.e., accuracy) as our primary metric, as it **directly reflects the performance gains in this single-pass scenario and ensures a fair comparison with relevant baselines**. **Furthermore, this choice aligns with a significant trend in the field: as the reasoning abilities of state-of-the-art models have become more powerful and reliable, technical reports from leading labs have increasingly emphasized pass@1 as a core metric for evaluating practical, out-of-the-box performance.**
>
> We fully share the reviewer's concern for robustness. **By averaging results over multiple runs, pass@1 can also effectively mitigate randomness and ensure stable conclusions.** To this end, we have already implemented rigorous validation procedures. As stated in our paper, we performed 4 independent runs for smaller datasets (like AMC and AIME) and reported the average. Furthermore, to demonstrate that our accuracy gains are statistically significant, we provide a detailed analysis in **Appendix K (Table 9)** with error bars and 95% confidence intervals based on up to 8 runs.
>
> **Results with Confidence Intervals**
>
>  | Model | GSM8K | MATH | AMC23 | AIME24 | GPQA |
>  |--------|-------|------|-------|--------|------|
>  | Vanilla (ds-7B) | 0.897 [0.891, 0.902] | 0.877 [0.869, 0.884] | 0.794 [0.767, 0.821] | 0.425 [0.400, 0.449] | 0.247 [0.203, 0.291] |
>  | DEER (ds-7B) | 0.904 [0.896, 0.912] | 0.897 [0.883, 0.911] | 0.856 [0.835, 0.878] | 0.492 [0.463, 0.520] | 0.299 [0.257, 0.341] |
>  | Vanilla (Qwen3-14B) | 0.948 [0.942, 0.955] | 0.938 [0.932, 0.943] | 0.938 [0.918, 0.957] | 0.708 [0.660, 0.757] | 0.596 [0.571, 0.621] |
>  | DEER (Qwen3-14B) | 0.955 [0.949, 0.962] | 0.942 [0.936, 0.948] | 0.953 [0.940, 0.967] | 0.754 [0.718, 0.791] | 0.587 [0.566, 0.608] |
>
> These results confirm that the improvements from DEER are consistent and not an artifact of randomness.

---

> > ### Author Response · Authors · 2025-11-19
> > **Response to Reviewer Ax2Q (2/2)**
> >
> > ## Response to weakness 3: Include more ablation studies
> >
> > We thank the reviewer for pushing us to conduct a more thorough ablation study. **While we had initial ablations in Section 4.3 (Table 5, 6), we have, based on your valuable feedback, conducted a new set of comprehensive ablation experiments during this rebuttal period.**
> >
> > We designed three variants to isolate the effects of our two core modules: the **Reasoning Transition Monitor** and the **Confidence Evaluator**.
> >
> > **DEER-Static**：Replaces dynamic confidence with a fixed heuristic (exit after 5 'Wait's)
> >
> > **DEER-Length**：Replaces transition monitor with a fixed length (exit check every 200 tokens)
> >
> > **DEER-\n\n**：Replaces transition monitor with a simpler heuristic (exit check at every \n\n)
> >
> > We conducted experiments on the DeepSeek-7B, Qwen3-14B, and Qwen3-8B models.
> >
> > Below are the summarized results on the DeepSeek-7B model：
> >
> >
> > |Method| Overall Acc| Overall CR|
> > |--|--|--|
> > |DEER-W (Full)| 69.2%| 61.5%|
> > |DEER-Static|  62.6% (-6.6)| 52.3%|
> > |DEER-Length |67.7% (-1.5)| 60.7%|
> > |DEER-\n\n| 68.8% (-0.4)| 57.0%|
> >
> >
> > These results clearly demonstrate:
> >
> > 1. **The necessity of the dynamic Confidence Evaluator**: Replacing it with a fixed heuristic (DEER-Static) leads to a catastrophic drop in accuracy (-6.6). Static rules fail to adapt to varying problem difficulties.
> >
> > 2. **The importance of the intelligent Reasoning Transition Monitor**: While simpler heuristics like \n\n also works, our proposed monitors achieve a better balance. Fixed-length truncation (DEER-Length) is suboptimal as it can abruptly cut off coherent thoughts. Moreover, the \n\n approach, while effective, incurs higher computational overhead due to excessive checks.
> >
> > We will add a new subsection to the final paper with these detailed results and analyses. Your suggestion has significantly strengthened our paper.
> >
> > The full results are presented below:
> >
> > **DeepSeek-R1-Distill-Qwen-7B**
> >
> >  | Method | GSM8K (Acc/Len/CR) | MATH (Acc/Len/CR) | AMC (Acc/Len/CR) | AIME (Acc/Len/CR) | GPQA (Acc/Len/CR) | Overall (Acc/CR) |
> >  |--|-----|-----|------|---|-----|-----|
> >  | Vanilla | 89.6 / 1484 / 100% | 87.4 / 3858 / 100% | 78.8 / 6792 / 100% | 41.7 / 13765 / 100% | 23.7 / 10247 / 100% | 64.2 / 100% |
> >  | DEER-W | 90.6 / 917 / 61.8% | 89.8 / 2143 / 55.5% | 85.0 / 4451 / 65.5% | 49.2 / 9839 / 71.5% | 31.3 / 5469 / 53.4% | 69.2 / 61.5% |
> >  | DEER-Ent | 90.8 / 876 / 59.0% | 89.2 / 2261 / 58.6% | 85.0 / 4072 / 60.0% | 48.4 / 8961 / 65.1% | 29.6 / 5037 / 49.2% | 68.6 / 58.4% |
> >   | DEER-\n\n | 90.9 / 854 / 57.5% | 89.4 / 2203 / 57.1% | 85.0 / 3985 / 58.7% | 48.4 / 8764 / 63.7% | 30.1 / 4912 / 47.9% | 68.8 / 57.0% |
> >  | DEER-Length | 90.5 / 908 / 61.2% | 88.8 / 2347 / 60.8% | 83.8 / 4256 / 62.7% | 46.7 / 9298 / 67.6% | 28.9 / 5234 / 51.1% | 67.7 / 60.7% |
> >  | DEER-Static | 89.8 / 1436 / 96.8% | 86.2 / 1821 / 47.2% | 72.5 / 2744 / 40.4% | 35.0 / 5643 / 41.0% | 29.4 / 3710 / 36.2% | 62.6 / 52.3% |
> >
> > **Qwen3-14B**
> >
> >  | Method | GSM8K (Acc/Len/CR) | MATH (Acc/Len/CR) | AMC (Acc/Len/CR) | AIME (Acc/Len/CR) | GPQA (Acc/Len/CR) | Overall (Acc/CR) |
> >  |-----|----|----|-----|------|--|----|
> >  | Vanilla | 95.1 / 2047 / 100% | 93.8 / 4508 / 100% | 95.0 / 7139 / 100% | 70.0 / 10859 / 100% | 60.1 / 7339 / 100% | 82.8 / 100% |
> >  | DEER-W | 95.3 / 840 / 41.0% | 94.0 / 3074 / 68.2% | 95.0 / 4763 / 66.7% | 76.7 / 7619 / 70.2% | 57.6 / 2898 / 39.5% | 83.7 / 57.1% |
> >  | DEER-Ent | 96.1 / 803 / 39.2% | 93.8 / 2979 / 66.1% | 93.3 / 4903 / 68.7% | 73.3 / 7128 / 65.6% | 58.1 / 2818 / 38.4% | 82.9 / 55.6% |
> >   | DEER-\n\n | 95.9 / 787 / 38.4% | 93.8 / 2924 / 64.9% | 93.8 / 4721 / 66.1% | 73.3 / 6987 / 64.3% | 58.3 / 2748 / 37.5% | 83.0 / 54.2% |
> >  | DEER-Length | 95.8 / 835 / 40.8% | 93.4 / 3093 / 68.6% | 92.5 / 5089 / 71.3% | 71.7 / 7391 / 68.1% | 57.3 / 2957 / 40.3% | 82.1 / 57.8% |
> >  | DEER-Static | 95.1 / 1955 / 95.5% | 90.8 / 2254 / 50.0% | 82.5 / 3356 / 47.0% | 43.3 / 4539 / 41.8% | 52.9 / 2201 / 30.0% | 72.9 / 52.9% |
> >
> > **Qwen3-8B**
> >
> >  | Method | GSM8K (Acc/Len/CR) | MATH (Acc/Len/CR) | AMC (Acc/Len/CR) | AIME (Acc/Len/CR) | GPQA (Acc/Len/CR) | Overall (Acc/CR) |
> >  |--------|------|----|----|------|----|------|
> >  | Vanilla | 94.9 / 2245 / 100% | 91.2 / 5216 / 100% | 87.5 / 7986 / 100% | 65.0 / 12110 / 100% | 51.5 / 9145 / 100% | 78.0 / 100% |
> >  | DEER-W | 95.2 / 1071 / 47.7% | 92.6 / 2732 / 52.4% | 92.5 / 4392 / 55.0% | 61.7 / 8796 / 72.6% | 52.5 / 3111 / 34.0% | 78.9 / 52.3% |
> >  | DEER-Ent | 95.8 / 1037 / 46.2% | 93.6 / 2789 / 53.5% | 91.3 / 4003 / 50.1% | 63.3 / 8328 / 68.8% | 51.5 / 3248 / 35.5% | 79.1 / 50.8% |
> >   | DEER-\n\n | 95.6 / 1012 / 45.1% | 93.4 / 2713 / 52.0% | 91.3 / 3895 / 48.8% | 63.3 / 8157 / 67.4% | 51.8 / 3146 / 34.4% | 79.1 / 49.5% |
> >  | DEER-Length | 95.5 / 1078 / 48.0% | 93.2 / 2896 / 55.5% | 90.0 / 4183 / 52.4% | 61.7 / 8651 / 71.4% | 50.8 / 3409 / 37.3% | 78.2 / 52.9% |
> >  | DEER-Static | 95.2 / 2095 / 93.3% | 89.4 / 2608 / 50.0% | 78.8 / 3194 / 40.0% | 36.7 / 5471 / 45.2% | 39.7 / 2743 / 30.0% | 68.0 / 51.7% |

---

### Official Review · Reviewer_gfWw · 2025-11-02

**Soundness:** 2
**Presentation:** 2
**Contribution:** 2
**Rating:** 4
**Confidence:** 3

**Summary:**

This paper proposes DEER, a training-free method for eliciting efficient reasoning from LRM by dynamically early-exiting CoT generation. The authors argue overthinking harms both latency and accuracy in long CoTs. Instead of using fixed heuristics, DEER watches for reasoning transitions (linguistic markers like “Wait” or entropy spikes), briefly induces a trial answer, and computes a geometric-mean confidence from token probabilities. If confidence exceeds a threshold, it stops thinking and outputs the answer. A robust variant, DEER-Pro, averages multiple inductions and MAD-calibrates confidence. Across 10 benchmarks and 11 models, DEER cuts reasoning length while achieving higher accuracy over the vanilla LRM.

**Strengths:**

1. The proposed method is a training-free and plug-and-play approach for eliciting efficient reasoning from LRMs, which has clear advantages over training-based methods.
2. The experiments demonstrate that the method improves efficiency and accuracy simultaneously.
3. The method is well-motivated and clearly presented.

**Weaknesses:**

1. Confidence may not always serve as a reliable signal for early exit. Prior work [1] has shown that explicitly encouraging reflection (such as by appending tokens like “wait”) can help the model identify and correct earlier mistakes. In contrast, DEER’s reliance on confidence may lead the model to terminate reasoning prematurely: it could be highly confident yet incorrect, losing the opportunity to self-correct in later reasoning steps.
2. It remains unclear how DEER and DEER-Pro achieve higher accuracy than the vanilla model in Table 1 despite substantial reductions in token length. Without a detailed explanation, it is difficult to justify the improvement.
3. The fairness of the reported comparisons is uncertain. DEER-Pro employs a parallel decoding strategy, which inherently introduces additional computational overhead compared to single-path decoding methods. This difference complicates claims of efficiency and may limit the validity of direct, apples-to-apples comparisons with baseline models.

References
1. s1: Simple test-time scaling

**Questions:**

Can this training-free method for efficient reasoning be adapted to be a training-based method to further improve efficiency and accuracy?

---

> ### Author Response · Authors · 2025-11-19
> **Response to Reviewer gfWw (1/3)**
>
> We sincerely thank you for your thoughtful and constructive review. We are encouraged that you recognized our method as a well-motivated, plug-and-play approach. Below, we address each of your concerns in detail.
>
> ## Response to weakness 1: More reflection (proposed by S1) vs. Early Exit (Ours)
>
> We appreciate you raising this crucial point. We address this concern from three perspectives(I, II, III):
>
> **I.** First, we would like to clarify that our work (DEER) and the cited work (s1) address two distinct, almost opposite, challenges faced by reasoning models at different capability levels. S1 focuses on tackling the **underthinking** problem in **_weaker models_** by encouraging more reflection. In contrast, DEER is designed to solve the **overthinking** problem prevalent in **_powerful Large Reasoning Models (LRMs)_**. Therefore, the finding that more reflection benefits weaker models does not contradict our conclusion that early exit is beneficial for stronger models.
>
> The success of the **budget forcing** strategy in s1 (their reflection encouraging method) relies on two specific conditions:
>
> - **The reasoning capability of the model adopted by S1  is relatively limited.** The s1-32B model mentioned in their paper was distilled on only 1k samples, resulting in relatively weak reasoning capabilities of this model. Besides, s1-32B's inherent tendency for reflection is not as pronounced as that of current state-of-the-art LRMs. Thus, **_it suffers more from underthinking than overthinking._**
>
> - **The token budget in s1's experiments is severely constrained.** As shown in Figure 4(a) of their paper, the thinking process is limited to a maximum of 4k tokens, which is often insufficient for complex reasoning tasks like AIME.
> In our work, however, we find that mainstream LRMs typically generate 10k-13k tokens or more to achieve their optimal performance on AIME24 (as shown in DEER's Table 1). Even after an early exit, DEER's reasoning sequences are still 8k-10k tokens long on average, far exceeding the 4k token limit in s1.
>
> To further validate this point, we will add a new experiment in the final version of our paper. We replicated the budget forcing strategy from on stronger models within our experimental setup. Our preliminary results indicate that under conditions of sufficient reasoning capability, forcing additional reflection steps fails to improve accuracy and even degrades it.  Besides, more reflection steps consistently increase inference latency. This strongly supports our core motivation that overthinking can be harmful.
>
> ### Preliminary Experimental Results on encouraging more reflections based on current SoTA LRMs
>
> **DeepSeek-R1-Distill-Qwen-7B**
>
>  | Method | GSM8K (Acc/Tok) | MATH-500 (Acc/Tok) | AMC23 (Acc/Tok) | AIME24 (Acc/Tok) | Overall (Acc/Tok) |
>  |--------|-----------------|---------------------|-----------------|------------------|------------------|
>  | Vanilla | 90.5 / 1608 | 92.0 / 3952 | 90.0 / 7039 | 55.0 / 10083 | 81.9 / 5671 |
>  | s1-2xWait | 90.2 / 2006 | 92.6 / 5078 | 90.0 / 7685 | 53.3 / 11301 | 81.5 / 6518 |
>  | s1-4xWait | 89.3 / 3218 | 91.0 / 6317 | 81.3 / 11337| 50.0 / 13682 | 77.9 / 8639 |
>  | s1-6xWait | 87.5 / 5051 | 89.6 / 7856 | 75.0 / 14835| 40.0 / 16743 | 73.0 / 11121 |
>
> **Qwen3-8B**
>
>  | Method | GSM8K (Acc/Tok) | MATH-500 (Acc/Tok) | AMC23 (Acc/Tok) | AIME24 (Acc/Tok)  | Overall (Acc/Tok) |
>  |--------|-----------------|---------------------|------------------|------------------|------------------|
>  | Vanilla | 95.7 / 2246 | 93.8 / 5368 | 93.8 / 9424 | 70.0 / 16717 | 88.3 / 8439 |
>  | s1-2xWait | 96.0 / 2677 | 93.4 / 6243 | 95.0 / 10892 | 70.0 / 18396 | 88.6 / 9552 |
>  | s1-4xWait | 95.4 / 3013 | 93.8 / 7350 | 93.8 / 12038 | 68.3 / 21207 | 87.8 / 10902 |
>  | s1-6xWait | 94.8 / 3596 | 93.4 / 8698 | 92.5 / 14697 | 66.6 / 24630 | 86.8 / 12905 |
>
>
>
>
> **II**. Second, our core motivation is that more thinking does not always lead to better results, particularly for the current state-of-the-art LRMs rather than the weaker models like s1. On the contrary, overthinking can degrade performance. **We provide direct evidence in Figure 1 (Questions 11, 19, and 26 on AIME24),** where the correct answers are exclusively obtained through early exit. **Continuing the reasoning process leads the model to change its correct answer to an incorrect one**. DEER's purpose is precisely to capture the "Pearl Reasoning" moment before the model's reasoning becomes derailed.

---

> > ### Author Response · Authors · 2025-11-19
> > **Response to Reviewer gfWw (2/3)**
> >
> > **III**. Finally, regarding your concern that relying on confidence might prematurely terminate reasoning and lose opportunities for self-correction, we acknowledge this as a potential issue and have provided extensive analysis in our paper.
> >
> > - **Macro-level behavior**: Our experiments (**Figure 8**) show that DEER behaves intelligently and cautiously. For difficult tasks like AIME, the early-exit rate naturally drops to around 20%, **indicating that the model avoids premature termination when deep reasoning is required**. Crucially, the accuracy of these early-exit samples is over 90%, significantly higher than the model's baseline accuracy on the task. This demonstrates that DEER's early-exit decisions are of high quality.
> >
> > - **Underlying mechanism**: This high-quality decision-making is supported by the **confidence polarization phenomenon** we identified (Appendix J, Table 12). Correct answers consistently exhibit high and stable confidence scores, while incorrect ones have significantly lower scores, creating a clear decision boundary. This provides a strong empirical foundation for using confidence as a reliable signal.
> >
> > - **Robustness enhancement**: To address potential instances of overconfident yet incorrect predictions, we designed **DEER-Pro**. Its core C_MAD penalty term essentially tests the stability and consistency of the model's confidence. An early exit is triggered only when the model demonstrates high confidence across multiple, varied prompts, which effectively mitigates the risks associated with relying on a single, potentially unreliable signal.
> >
> >
> > ## Response to weakness 2: how DEER and DEER-Pro achieve higher accuracy than Vanilla CoT.
> >
> > We appreciate the request for a clearer explanation, and we are confident that our paper provides a comprehensive analysis to support this finding. **For current LRMs, the benefits of addressing overthinking far outweigh the negative impacts introduced by early exit.**
> >
> > First, we provide strong **qualitative evidence**. Our pilot experiments in **Figure 1 (Questions 11, 19, 26)** and **Figure 2(b)** already establish that overthinking is a genuine issue.  The primary source of DEER's accuracy gain is its success in **preventing the model from invalidating its own correct, intermediate conclusions through excessive reasoning**. This is further illustrated in our case studies (**Figures 13-16**), where Vanilla CoT finds the correct path early on but then gets lost in unnecessary self-doubt and redundant verification loops, while DEER preserves the correct result by exiting at the opportune moment.
> >
> > Furthermore, we offer detailed **quantitative analysis**. The accuracy improvement is systemic, not coincidental. The analysis in **Figure 11** clearly shows that **DEER corrects more answers (green bars) than it alters incorrectly (red bars)** through early exits. This indicates that the early-exit mechanism provides a positive impact on accuracy. This finding, combined with the high accuracy of early-exited samples shown in **Figure 8(b)**, collectively explains how DEER achieves a simultaneous improvement in accuracy and a substantial reduction in sequence length.

---

> ### Author Response · Authors · 2025-11-19
> **Response to Reviewer gfWw (3/3)**
>
> ## Response to weakness 3: The fairness of the reported comparisons
>
> We agree that fair comparison is paramount and would like to clarify any potential misunderstandings.
>
> First, it is essential to distinguish between our two proposed methods. Our core method, **DEER, is a single-path decoding approach**. Its comparison with Vanilla CoT and other baselines in Table 1 is a **direct and fair, apples-to-apples comparison**. The results demonstrate that even this base version of DEER achieves significant improvements in both efficiency and accuracy.
>
> Regarding the enhanced version, DEER-Pro, while it does introduce parallel answer induction, the associated overhead is negligible compared to the computational savings from reducing the CoT length. The extra computation occurs only during the brief answer generation phase. Both our theoretical analysis in **Appendix H** and the empirical statistics (provided below, showing the number of early-exit attempts, k) confirm that this overhead is minimal. For instance, **with N=4 and an average answer length of ~20 tokens**, the computational cost is far less than that of generating a reasoning chain that is thousands of tokens long.
>
> | Dataset       | AIME24 | AMC23 | MATH | GPQA | GSM8K |
> |---------------|-----|-------|----------|-----|------|
> | Qwen3-14B | 8.6  |  6.1  | 3.608  | 2.162  | 2.253 |
> | Qwen3-8B | 8.93  |  6.675  | 3.92  | 2.561  | 2.467 |
>
> Finally, the ultimate measure of efficiency is end-to-end latency. As we demonstrate in Figure 7, the branch-parallel decoding acceleration strategy (described in Section 3.3) can be applied to DEER-Pro. The results show that **the wall-clock time saved by reducing the extensive CoT generation far outweighs the minor overhead introduced by the confidence evaluation mechanism**. This confirms that our claims of efficiency improvement are robust and well-supported by empirical evidence.
>
> ## Response to question 1: Can this training-free method be adapted to be a training-based method?
>
> This is an excellent and insightful question that touches upon the natural extension of our work. We fully agree that adapting this training-free early exit mechanism into a training-based framework is a highly promising direction, and we have indeed already designed and experimentally validated such an approach.
> **However, the scope of the current paper is intentionally focused on delivering a plug-and-play solution that can be seamlessly integrated into any existing LRM without the need for costly training**. For this reason, we decided that the training-based variant, which constitutes a significant research effort in its own right, would be best presented as an independent work in the future. We are happy to share the core concepts of our design to illustrate the significant potential of this direction. We believe a particularly powerful paradigm for this is **Reinforcement Learning (RL)**, built on the following ideas:
>
> - **An Innovative Sampling Strategy**: Our training-based early exit paradigm would first generate a single, complete Chain-of-Thought  path. Then, multiple early-exit points would be serially selected along this one path to form a sample group of trajectories with varying reasoning depths. This allows for a more direct comparison of different thought lengths on the exact same prefix of reasoning.
>
> - **A Refined Reward Mechanism**: Within the sample group described above, the shortest trajectory that yields a correct answer would receive the highest reward. Subsequent trajectories that are longer but also correct would receive progressively smaller rewards. This reward structure would precisely incentivize the model to terminate its reasoning process as soon as its thoughts are sufficient for a correct answer.
>
> We tested this approach with the following results (max_len: 32k, temperature=0.6, top_p=0.95), showing that this training framework can significantly enhance both model efficiency and accuracy.
>
> **DeepSeek-R1-Distill-Qwen-7B**
>
>  | Method | GSM8K (Acc/Tok) | AIME24 (Acc/Tok) | AMC23 (Acc/Tok) | MATH-500 (Acc/Tok) | GPQA (Acc/Tok) | Overall (Acc/Tok) |
>  |--------|------|--------|---------|-----|-----|------|
>  | Vanilla | 92.4 / 1833 | 55.4 / 13232 | 77.2 / 9693 | 85.8 / 5590 | 50.1 / 15385 | 70.02 / 9147 |
>  | DEER | 88.8 / 917 | 53.3 / 10971 | 87.5 / 4142 | 91.8 / 2431 | 47.5 / 5280 | 73.78 / 4748 |
>  | training-based | 93.8 / 906 | 56.0 / 7377 | 87.5 / 3494 | 92.4 / 2252 | 50.8 / 3751 | 76.10 / 3556 |
>
> **DeepSeek-R1-Distill-Qwen-14B**
>
>  | Method | GSM8K (Acc/Tok) | AIME24 (Acc/Tok) | AMC23 (Acc/Tok) | MATH-500 (Acc/Tok) | GPQA (Acc/Tok) | Overall (Acc/Tok) |
>  |--------|-----|--------|-----|------|----|----|
>  | Vanilla | 94.2 / 2129 | 64.4 / 11099 | 90.5 / 5527 | 93.5 / 3844 | 59.2 / 6034 | 80.36 / 5727 |
>  | DEER | 93.3 / 982 | 70.0 / 10335 | 90.0 / 4349 | 91.4 / 2753 | 57.1 / 4767 | 80.36 / 4637 |
>  | training-based | 96.2 / 724 | 64.4 / 6712 | 91.9 / 3352 | 93.6 / 2146 | 59.3 / 3334 | 81.08 / 3253 |

---

> > ### Comment · Reviewer_gfWw · 2025-11-26
> >
> > I thank the authors for providing clear justifications. My concerns are mostly addressed and I have revised the score accordingly.

---

> > > ### Author Response · Authors · 2025-11-26
> > >
> > > We sincerely thank you for the positive feedback and for acknowledging our clarifications. We are pleased that our responses have addressed your concerns.

---

### Official Review · Reviewer_LEHd · 2025-11-03

**Soundness:** 3
**Presentation:** 3
**Contribution:** 3
**Rating:** 6
**Confidence:** 4

**Summary:**

Large reasoning models often produce excessively long reasoning sequences, leading to computational inefficiency. To address this issue, the paper proposes DEER, a method that detects transition points during model reasoning, based on linguistic markers or entropy, and prompts the model to generate trial answers at these points. A confidence evaluation mechanism is then applied to decide whether to stop further reasoning and produce a final answer or continue generating additional thoughts.

**Strengths:**

-	The proposed method is simple and effectively reduces the number of output tokens in practice.
-	The experiments are comprehensive, covering a wide range of models (DeepSeek-R1-Distill-Qwen series, Qwen3 series, QwQ-32B, and DeepSeek-R1) across 10 benchmarks.

**Weaknesses:**

-	More details about the branch-parallel acceleration strategy (Section 3.3) are needed. The current description lacks clarity.
-	The technical novelty is relatively limited. Using linguistic markers or entropy to identify reasoning transitions is not entirely new, as similar ideas have been well explored in prior and concurrent works [1-6]. It would be beneficial to include a discussion of these related approaches.
-	The paper lacks an evaluation of the quality of reasoning steps. It remains unclear whether early exits lead to oversimplified reasoning or disrupt the reasoning flow. A more detailed quality analysis would strengthen the work.

[1] Yan, Hang, et al. "Mur: Momentum uncertainty guided reasoning for large language models." arXiv preprint arXiv:2507.14958 (2025).
[2] Wang, Shenzhi, et al. "Beyond the 80/20 rule: High-entropy minority tokens drive effective reinforcement learning for llm reasoning." arXiv preprint arXiv:2506.01939 (2025).
[3] Fu, Yichao, et al. "Efficiently Scaling LLM Reasoning with Certaindex." arXiv preprint arXiv:2412.20993 (2024).
[4] Agarwal, Shivam, et al. "The unreasonable effectiveness of entropy minimization in llm reasoning." arXiv preprint arXiv:2505.15134 (2025).
[5] Huang, Jiameng, et al. "Efficient reasoning for large reasoning language models via certainty-guided reflection suppression." arXiv preprint arXiv:2508.05337 (2025).
[6] Vanhoyweghen, A., Verbeken, B., Algaba, A., & Ginis, V. (2025). Lexical hints of accuracy in llm reasoning chains. arXiv preprint arXiv:2508.15842.

**Questions:**

n/a

---

> ### Author Response · Authors · 2025-11-19
> **Response to Reviewer LEHd (1/3)**
>
> Thank you for your detailed review and constructive feedback. We are grateful for your positive assessment of our method's **simplicity and effectiveness** in token reduction, as well as the **comprehensiveness** of our experiments. Your recognition of these strengths is very encouraging.
> We have carefully considered the weaknesses you pointed out and would like to address your concerns as follows.
>
> ## Response to weakness 1: More details about the branch-parallel acceleration strategy
>
> Thank you for pointing out that the description of this part in the current version is indeed too concise. The implementation of branch-parallel acceleration is illustrated in **Figure 9** in the appendix. We will add more detailed textual descriptions in the final version to elaborate on the implementation of this acceleration mechanism.
>
> The core idea of the Branch-Parallel acceleration strategy is to parallelize the two branches (trial answer generation and continued reasoning) instead of executing them serially when a reasoning transition point is detected. As illustrated in Figure 9, upon encountering a reasoning transition point (e.g., a "Wait" token), we construct a unified sequence containing both branches:
>
> [Shared Context] → [Branch 1: Continue Reasoning] || [Branch 2: Trial Answer]
>
> The key innovation lies in a **specialized causal attention mask** that enables both branches to be generated simultaneously while maintaining their independence:
>
> - Shared region (tokens 1 to t): Full causal visibility for all subsequent tokens.
> - Branch 1 region (tokens t+1 to t+n₁): Causal mask within this branch, with no visibility to Branch 2.
> - Branch 2 region (tokens t+n₁+1 to t+n₁+n₂): Causal mask within this branch, with no visibility to Branch 1.
>
> This mask ensures that while both branches share the same computational graph and can be processed in a single forward pass, they remain causally independent after the branching point.
>
>  To prevent memory explosion from maintaining multiple branches, we implement confidence-driven cache pruning. During parallel generation, we maintain separate KV cache partitions for each branch. Based on the confidence evaluation, we prune the cache by discarding the KV cache of the continued reasoning branch upon a confirmed early exit, or that of the trial answer branch otherwise.
>
> **Efficiency Analysis.** Let s denote the shared context length, and b₁, b₂ denote the branch length. The computational complexity comparison is:
>
> - Serial DEER: O(s) + O(b₂) + O(s) + O(b₁) = O(2s + b₁ + b₂)
> - Branch-Parallel DEER: O(s + max(b₁, b₂))
>
> The parallel approach eliminates redundant computation of the shared context and overlaps the generation of the branches.
>
> **Implementation Considerations.** While our attention-mask approach is theoretically optimal, we acknowledge that practitioners may prefer simpler alternatives. Two independent forward passes, though less efficient (requiring duplicate KV cache computation for the shared context), offer easier implementation and broader framework compatibility.

---

> > ### Author Response · Authors · 2025-11-19
> > **Response to Reviewer LEHd (2/3)**
> >
> > ## Response to weakness 2: Discussion of more related works
> >
> > Thank you for pointing out these important contemporaneous works. **A deeper comparison has further strengthened our confidence in DEER's unique methodology.** We summarize the comparison as follows:
> >
> > - **Works [2] and [6]** focus primarily on analysis. [2] identifies lexical cues in reasoning chains, while [6] reveals the importance of high-entropy tokens at decision points. These studies provide valuable insights into why certain signals are effective. DEER's contribution lies in building a complete system that integrates monitoring, inducing, and evaluating, effectively translating these signals into reliable decisions.
> > - **Work [4]** investigates the benefits of entropy minimization across fine-tuning, RL, and inference. While we share the use of entropy as a concept, DEER does not treat entropy minimization as its primary research focus. Instead, we employ entropy as one of several auxiliary methods for identifying potential exit points.
> > - **Work [1]** dynamically allocates thinking budgets for complex multi-path sampling processes, which requires computationally expensive uncertainty calculations. In contrast, DEER targets the more common single-path generation scenario, providing a lightweight and efficient early exit mechanism.
> > - **Work [3]** primarily addresses system-level token budget management. We noted that their revised preprint (v2, 2025.05.27) incorporates the "Probe-In-The-Middle" technique, which is essentially the Dynasor-CoT method we use as a baseline. This approach requires probing at fixed token intervals, which can disrupt the natural flow of reasoning. DEER intervenes at natural transition points, better preserving reasoning coherence. This distinction also addresses your third concern. In addition, DEER's criteria for exiting reasoning and DEER-Pro's unique robust design are both significantly different from them.
> > - **Work [5]** reduces overthinking indirectly by suppressing reflection triggers. DEER's approach is more direct and robust in comparison.
> >
> > DEER's novelty lies not in any single technical element but in its systematic design and integration. We acknowledge these important studies as valuable contemporaneous works, and their emergence collectively demonstrates the significance of this research direction. Given the substantial volume of concurrent research, we initially focused on the most directly relevant works. We will expand our discussion in the final version to include these additional studies, better positioning DEER's unique contribution within this evolving research landscape. **As these are concurrent preprint publications, we believe they complement rather than diminish the novelty of our method.**
> >
> >
> > [1] Yan, Hang, et al. "Mur: Momentum uncertainty guided reasoning for large language models." arXiv preprint arXiv:2507.14958 (2025).
> >
> > [2] Wang, Shenzhi, et al. "Beyond the 80/20 rule: High-entropy minority tokens drive effective reinforcement learning for llm reasoning." arXiv preprint arXiv:2506.01939 (2025).
> >
> > [3] Fu, Yichao, et al. "Efficiently Scaling LLM Reasoning with Certaindex." arXiv preprint arXiv:2412.20993 (2024).
> >
> > [4] Agarwal, Shivam, et al. "The unreasonable effectiveness of entropy minimization in llm reasoning." arXiv preprint arXiv:2505.15134 (2025).
> >
> > [5] Huang, Jiameng, et al. "Efficient reasoning for large reasoning language models via certainty-guided reflection suppression." arXiv preprint arXiv:2508.05337 (2025).
> >
> > [6] Vanhoyweghen, A., Verbeken, B., Algaba, A., & Ginis, V. (2025). Lexical hints of accuracy in llm reasoning chains. arXiv preprint arXiv:2508.15842.

---

> ### Author Response · Authors · 2025-11-19
> **Response to Reviewer LEHd (3/3)**
>
> ## Response to weakness 3: Evaluation of the quality of reasoning steps
>
> Thank you for this insightful question, which is crucial for evaluating DEER comprehensively. We conducted extensive qualitative and quantitative analyses to address this concern systematically. The results strongly support that DEER preserves and even enhances reasoning quality. While our appendix (**Figures 13-16**) already demonstrates how DEER prevents errors from overthinking, we have performed additional detailed analyses that will be included in the final version:
>
> ### 1. Qualitative Case Studies
>
> We analyzed typical cases across different scenarios to understand DEER's behavior:
>
> - **Case 1: Efficient and Accurate (Both DEER and Vanilla Correct)**: When both methods reached the correct answer, DEER produced significantly shorter reasoning chains while retaining all essential logical steps. It only omitted redundant verification stages. LLM-as-a-judge evaluations confirmed the completeness of DEER's reasoning, demonstrating that it enhances efficiency without sacrificing quality.
>
> - **Case 2: Failure Attribution Analysis (DEER Incorrect, Vanilla Correct)**: We candidly analyzed cases where DEER failed while the vanilla model succeeded. Our investigation revealed that these failures were not caused by premature termination leading to incomplete reasoning. Instead, they stemmed from the inherent stochasticity of the model's reasoning process on challenging problems. In these instances, DEER's reasoning path began with an incorrect premise from the outset. A control experiment confirmed this: forcing the model to continue reasoning from the exit point still resulted in an incorrect answer. This crucial finding demonstrates that DEER did not interrupt a promising path that could have self-corrected.
>
> Due to the word limit of the rebuttal, we will include Case 1 and Case 2 in the appendix of the paper (figure 19 on page 42, and figure 21 on page 43).
>
> ### 2. Quantitative Quality Assessment
>
> Beyond individual cases, we conducted a large-scale assessment using the **LLM-as-a-judge** paradigm. We randomly sampled 100 DEER responses from the MATH-500 dataset and used GPT-5 to evaluate reasoning completeness:
>
> - **97% of responses were judged as logically complete**
> - The 3 responses deemed incomplete actually contained minor logical leaps rather than missing critical steps. Even under strict evaluation, DEER's reasoning chains were structurally complete and sufficient to support the final answers.
>
> This comprehensive analysis, combining detailed case studies with broad quantitative evaluation, demonstrates that DEER effectively maintains reasoning quality and completeness while significantly improving efficiency.
>
> **LLM-as-a-judge prompt:**
>
> ```
> ## Role:
> You are an expert in mathematical and logical reasoning evaluation. Your task is to act as an impartial judge, assessing the quality and completeness of a reasoning process.
>
> ## Context:
> You will be given a problem, a sequence of reasoning steps produced by a student model, and the final answer derived from these steps.
>
> ## Task:
> Your evaluation should be based **ONLY** on the provided "Reasoning Steps". Do not solve the problem yourself or use your own knowledge. Your goal is to determine if the given reasoning steps logically and sufficiently support the given "Final Answer".
>
> ## Input:
> ### Question:
> {}
> ### Reasoning Steps:
> {}
> ### Final Answer:
> {}
> ## Evaluation Criteria & Output Format:
> Please provide your evaluation in a structured JSON format. Answer the following questions:
> {
>   "reasoning_sufficiency": "Sufficient" | "Insufficient",
>   "justification": "Provide a brief, step-by-step analysis explaining your ratings. If reasoning is insufficient, specify what key steps or logic are missing. If it is sufficient, confirm that all necessary components are present.",
> }
> ```
>
>
> Due to the word limit of the rebuttal, we will include the LLM-as-a-judge evaluation cases in the appendix of the paper (figure 20 on page 42).

---

### Author Response · Authors · 2025-11-24
**Authors' Official Comment**

Dear Reviewers,

Thank you again for your thoughtful and constructive reviews. We hope our responses have helped clarify the concerns raised, and we truly appreciate the opportunity to improve our work.

If there are any remaining questions or suggestions, we would be grateful to hear them. Your feedback is incredibly valuable to us.

Warm regards,

The Authors

---

### Author Response · Authors · 2025-12-01
**Summary of Rebuttal and Discussion for the Area Chair**

Dear Area Chair,

We hope this message finds you well. We are writing to provide a summary of our rebuttal and the subsequent discussion for our submission, "DYNAMIC EARLY EXIT IN REASONING MODELS". We understand the challenging circumstances following the recent data leak and appreciate the immense effort you are putting into the meta-review process.

First and foremost, to assist your evaluation, we wish to highlight a crucial fact from the discussion history: **prior to the score rollback, our paper's scores had been updated to 6, 6, 6.** Most notably, Reviewer gfWw, who initially assigned a score of 4, raised their rating to 6. The timestamp for this final score update is **November 26th, which predates the official ICLR notification regarding the data leak.** This timeline confirms that the reviewer's revised assessment was based solely on the scientific merits of our rebuttal and the subsequent discussion, free from any influence of the unfortunate events that followed.

There was a strong consensus among all three reviewers on our paper's core strengths. They praised our work for its:
- **Clear Motivation & Insightful Observation**: Acknowledging that we convincingly identify the critical "overthinking" problem in modern LRMs.
- **Simple Yet Effective Approach**: Highlighting DEER's design as a training-free, plug-and-play method that is easy to integrate, the effectiveness of which has been validated across 11 different model series.
- **Comprehensive Experiments & Strong Results**: Recognizing that our experiments are extensive and demonstrate simultaneous improvements in both efficiency and accuracy.

The reviewers' concerns were constructive, with **no fatal flaws identified**. We successfully addressed all points, further strengthening the paper:

1. **Resolved Core Concern of Reviewer gfWw (Score from 4 to 6):**
- The Key Question: The reviewer questioned why Early Exit is beneficial when prior work suggests More Reflection helps models self-correct.
- Our Resolution: We resolved this by showing they are solutions for two different problems. With new supporting experiments, we demonstrated that "More Reflection" helps weaker models overcome "under-thinking," while our "Early Exit" helps powerful LRMs avoid performance degradation from "overthinking." The reviewer confirmed their concerns were addressed and raised their score.

2. **Addressed Incremental Feedback from Reviewers LEHd & Ax2Q (Scores: 6):**
- Key Feedback: Reviewers requested deeper analysis on novelty, reasoning quality, and component contributions.
- Our Resolution: We provided comprehensive responses and strengthened the paper significantly by conducting three sets of extensive new experiments:
  - A large-scale LLM-as-a-judge evaluation (97% of responses judged logically complete), confirming reasoning quality is preserved.
  - Comprehensive new ablation studies isolating the impact of our core modules.
  - A new comparative experiment (replicating the "More Reflection" strategy) to support our core motivation.

The full context for this summary is available in the discussion threads for each reviewer. **We highly recommend reviewing these detailed exchanges**, as they powerfully illustrate how the rebuttal process was highly successful, leading to a consensus of acceptance and a substantially improved paper. We are confident it meets the high standards of ICLR.

Thank you for your time and careful consideration.

Sincerely,

The Authors of Submission #11793

---

### Meta-Review · Area_Chair_Y1Zn · 2025-12-05

**Summary:**

This paper proposes a method that does early exit in long reasoning models. The method works by looking at reasoning transition points in the reasoning model, then try to induce the answer at those points, followed by confidence evaluation. If the confidence of an induced answer at a reasoning transition point exceeds a threshold then the reasoning is early terminated and a final answer is generated. The authors evaluated on open source reasoning models of various sizes and demonstrate the proposed approach can achieve comparable performance as compared to the reasoning model with reduced token usage.

Reviewer concerns on clarity and ablations are mostly addressed by the authors by supplementing corresponding results.

**Reviewer Concerns:**

Details on branch-parallel strategy.
- The authors have provided illustration and added additional text into the appendix

Qualitative evaluation
- The authors have added evaluation

Comparison with s1
- Author clarified different target, added additional experiments

Abalation studies
- Authors added experiments on the ablations

Overall, most of the concerns are addressed by the authors

**Reviewer Scores:**

LEHd: 6
gfWw: 4 -> 6, the reviewer stated the questions got clarified and will raise the score
Ax2Q: 6

---

### Decision · Program_Chairs · 2026-01-26

Accept (Poster)